# Lamprophyre as the Source of Zircon in the Veneto Region, Italy

**Daria Zaccaria [1], Noemi Vicentini [1], Maria Grazia Perna [1],\*, Gianluigi Rosatelli [1], Victor V. Sharygin [2], Emma Humphreys-Williams [3], Will Brownscombe [3] and Francesco Stoppa [1]**

1 DiSPUTer, University "G. d'Annunzio" Chieti-Pescara, 66100 Chieti, Italy; daria.zacca@hotmail.it (D.Z.); noemi.vicentini@unich.it (N.V.); grosatelli@unich.it (G.R.); fstoppa@unich.it (F.S.)
2 VS Sobolev Institute of Geology and Mineralogy, Siberian Branch of the RAS, 630090 Novosibirsk, Russia; sharygin@igm.nsc.ru
3 The Natural History Museum, London SW7 5BD, UK; e.williams@nhm.ac.uk (E.H.-W.); w.brownscombe@nhm.ac.uk (W.B.)
\* Correspondence: mariagrazia.perna@unich.it; Tel.: +39-0871-3556490

**Abstract:** Discrete zircons, up to 9 mm in length, occur in alluvial deposits from the Veneto area. They are likely derived from the disaggregation of lamprophyric rocks belonging to a regional, pervasive dyke-swarm. Zircon and REE phases occur in both alkaline lamprophyres and connate calcite-bearing felsic lithics and their debris in lamprophyre breccia. We present 36 new complete U–Pb and trace element analyses of zircons and associated inclusions. We used a statistical approach on a larger dataset using new and literature data to evaluate the confidence figure to give an estimation of age of zircons. Inclusions suggest a genetic link with an $S–CO_2–ZrO–BaO–SrO–CaO$-rich fluid/melt possibly associated with carbonate-rich alkaline parental magma and a metasomatised mantle source. This paper confirms the importance of calcite–syenite and lamprophyre genetic link and zircon magmatic origin, in contrast with hydrothermal and metamorphic zircons. U–Pb dating by LA-ICP-MS provides time constrains (40.5–48.4 Ma, Lutetian), consistent with the age of the alkaline magmatic event. Trace element data indicate a link to anorogenic magmatism associated with mantle upwelling. Complex zoning is highlighted by cathodoluminescence images. The Veneto zircons are helpful for regional geological information and may have commercial potential as a critical resource for green technologies.

**Keywords:** zircon; lamprophyres; U–Pb dating; zircon geochemistry; Veneto region; Italy

## 1. Introduction

Zircon ($ZrSiO_4$) is an accessory mineral in many igneous and metamorphic rocks. It is chemically and physically resistant, has a high specific density and, thus, concentrates in so-called heavy-mineral sands (HMS). It may contain low content of Ti, Nb, and Pb, and relatively high Y and REE content (Y >1000 ppm, LREE <100 ppm and HREE ~800 ppm), which often form discrete phases as inclusions. Zircon has many applications in modern industry. For instance, plasma-spray technologies use it as a coating for airborne panels and components [1]. Yttrium extracted from zircons is a strengthening element in many materials and metal alloys [2]. Zirconium is part of the strategic-raw materials, which becomes extremely important for the electronic and digital supply chain and the high-tech industry and green technologies [3]. The global demand for zirconium is currently 120,000 t/year, with a steady growth expected annually. The growing global demand for critical raw materials (CRMs) generated by international markets is strongly unbalanced with the availability of CRMs in mineral deposits. The zirconium market was valued at USD 5.14 billion in 2019 and will reach USD 7.2 billion by 2026, expanding at a compound annual growth rate of 5.2% during the forecast period 2020–2026. The market growth relates to the increasing number of nuclear power plants in emerging economies such as India and China. Europe is facing the so-called "balance problem", so it is essential to look for new deposits and know their exploitability in the EU countries.

Zircon is considered a powerful tool for geochronology and also provided other information about genetic conditions and geodynamic associations. However, interpreting its geochemistry is complex and may be complicated and consequently requires a thoughtful approach, as suggested in this paper.

The zircons analyzed in this study occur in superficial soils at Le Fosse di Novale, Lonedo and other localities of the Vicentine province (Veneto Region, NE Italy) (Figure 1). Zircons in superficial soils are a promising marker for deeper zircon-ilmenite rich HMS. However, exploration is costly and challenging in a populated area. Therefore, we aim to constrain zircon source and transport would make more feasible a future exploration.

Due to their density, heavy minerals concentrate by the tractive forces operated by river water. Ilmenite/spinel (4.5–5 $\rho$), olivine (3.28–3.48 $\rho$), garnet (3.62–3.87 $\rho$), clinopyroxene (3.22–3.38 $\rho$), zircon (4.60–4.70 $\rho$) are present in the following relative percentage among heavy minerals in the Vicentine soils at 68%, 20%, 5%, 3.4%, and 3.4%, respectively. Of particular importance is the presence of the pyrope-rich garnet ($Pyr_{57}Alm_{29}Gro_7And_7Sps_1$) and Mg-rich ilmenite ($Ilm_{72}Gk_{27}Pr_1$, FeO—42.8–45.6, MgO—4.6–7.0, MnO—0.3–0.6 wt %), associated with zircon.

The potential source of zircons is a dike–diatreme system upstream of the zircon deposits [4,5]. Mafic alkaline igneous rocks are widespread in the Plateau of the Sette Comuni, Tonezza del Cimone, Castelletto di Rotzo and other localities in the vicinity of Pedescala in the Val d'Astico. These rocks are poorly studied and range in composition from lamprophyres (Castelletto di Rotzo) to olivine foidite with lamprophyres' affinity (Tonezza del Cimone). This work provides for a geochemical study, including U–Pb dating, to understand the relationship between the zircons in the alluvial deposits and those in the lamprophyre rocks. In particular, we studied the distribution of high field strength elements (HFSE) and rare earth element (REE) concentrations to distinguish magmatic, metamorphic (crust–mantle), and hydrothermal/metasomatic/metamictic zircons [6]. We statistically compared the ages determined in this study with previously published geochronology data for this area. In addition, we dated the Lonedo zircons, which were not considered in literature so far.

## 2. Geological Setting

The study area is a wedge of mostly undeformed Adria plate foreland between western and eastern south-verging South Alpine thrust systems, much to the south of the Peri Adriatic Line, a geo-suture division between Austro-Alpine and South-Alpine domains. South-Alpine tectonics is still active today in the study area, producing several historical earthquakes in the southern Alpine sector (e.g., Asolo Earthquake, 1965 M. 6.4) [7,8] (Figure 1). The South-Alpine domain is a 10 to 15 km thick retro-wedge consisting of upper crustal-slices, resting on the Adria-plate middle and lower crust. Miocene-Pliocene tectonics of the South-Alpine sector is coeval with the Tyrrhenian Sea opening [9] and postdates the initial magmatic activity on the Adria plate margin (Middle Eocene-Miocene). Intense sub-aerial erosion deeply affected the sedimentary and igneous rocks exposed post-Upper Miocene [10]. Veneto's magmatic events, both effusive and explosive, began with a pervasive alkaline lamprophyric dyke and diatreme swarm in Middle Eocene time. Intense regional thermometamorphism (brucite marbles) accompanied by high temperature limburgite submarine flow of fluid lavas associated with an intense heat flow along the Periadriatic Line in Oligocene time [11]. The sub-aerial activity consisted of the emission of large quantities of pyroclastic products, near-surface hyaloclastite deltas, and sub-aerial monogenic volcanoes related to a transtensive fault system NNW–SSE (Schio-Vicenza strike-slip system) [7], forming the volcanic districts of the Marostica, Lessini, Berici and Euganei Hills. Middle Oligocene volcanism produced vast deposits of layered tuffs, as well as lava flows [12]. Lately, the emission of viscous rhyolitic, trachytic and latitic volcanics may represent the final melting of crustal rocks due to ultrabasic melt underplating the Moho [12]. The Veneto volcanic districts were not affected by deformation located south of the Thiene–Bassano thrust and are in the Adria foreland (Figure 1). Tonezza, Castelletto

di Rotzo and Marostica Hills are deformed and located north of the more external South-Alpine thrust [11]. In the Veneto region, the crust is thin, so there was an upwelling of the mantle possibly due to a local hot spot, testified by P-wave seismic tomography, showing low-velocity anomalies at depth [12].

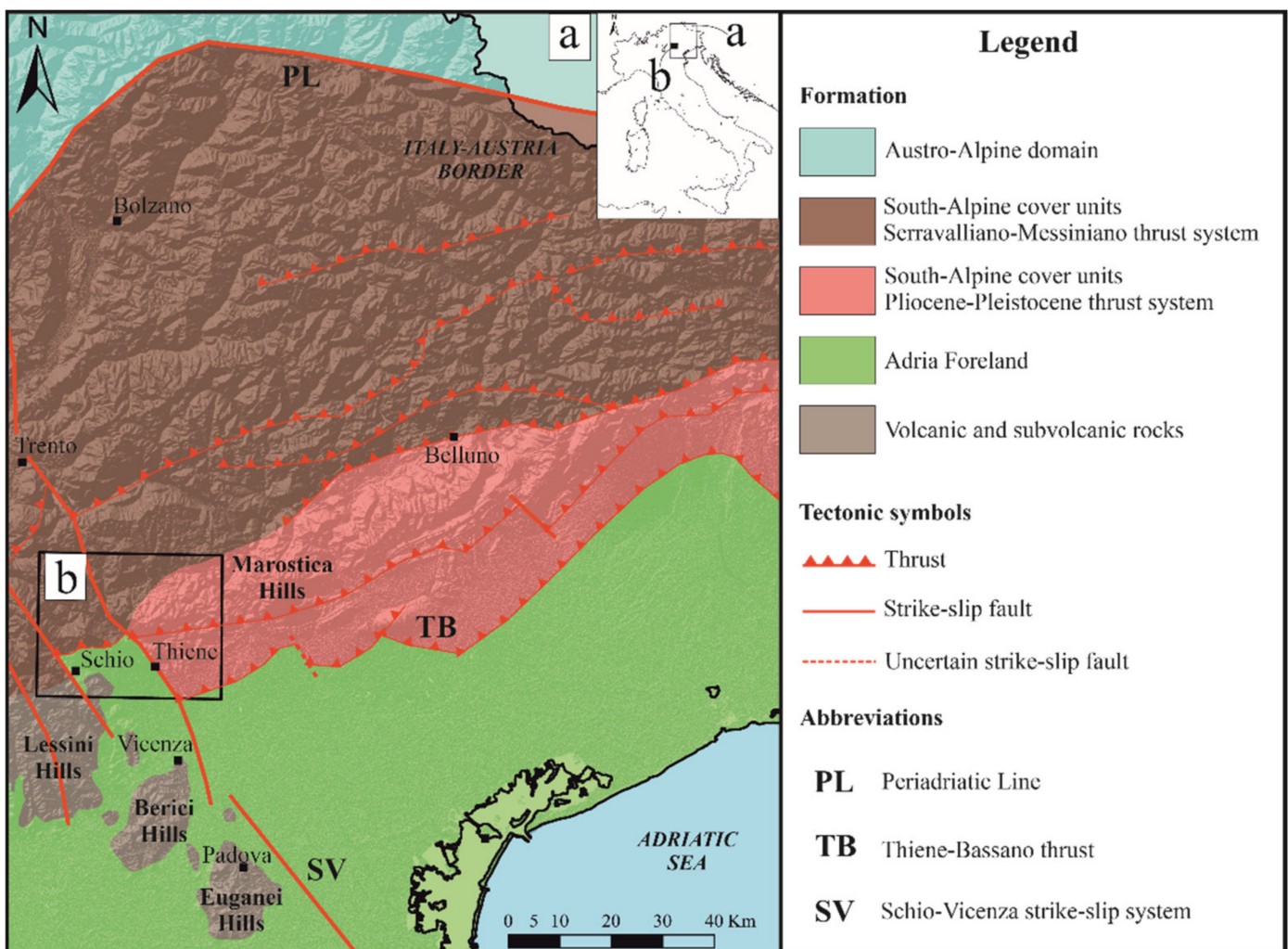

**Figure 1.** (**a**) Structural model of the Veneto area [13–15]; (**b**) study area corresponding to Figure 2.

Sub-volcanic dykes and diatremes at Tonezza del Cimone and Castelletto di Rotzo have an Oligocene age [16]. The entire area hosts dozens of dissected, en-echelon dykes varying from a meter to 10 m thick. Castelletto di Rotzo outcrop consists of a dyke-diatreme system, hosted in Barremian limestones (Majolica or Biancone) and maybe in Hettangian-Domerian limestones (Calcari Grigi group) and partially covered by fluvioglacial-morainic deposits (Figure 2) [17]. Tonezza del Cimone are poorly studied and deserve more attention and a future petrological study. Interestingly, one of the authors (FS) has discovered carbonatitic diatremes in Tonezza del Cimone, ultramafic rocks with CaO >30 wt% and $CO_2$ >20 wt % have recently been discovered (F. Stoppa personal communication). Mantle-xenoliths are frequent and testify to a rapid rise of the magma from the mantle towards the surface.

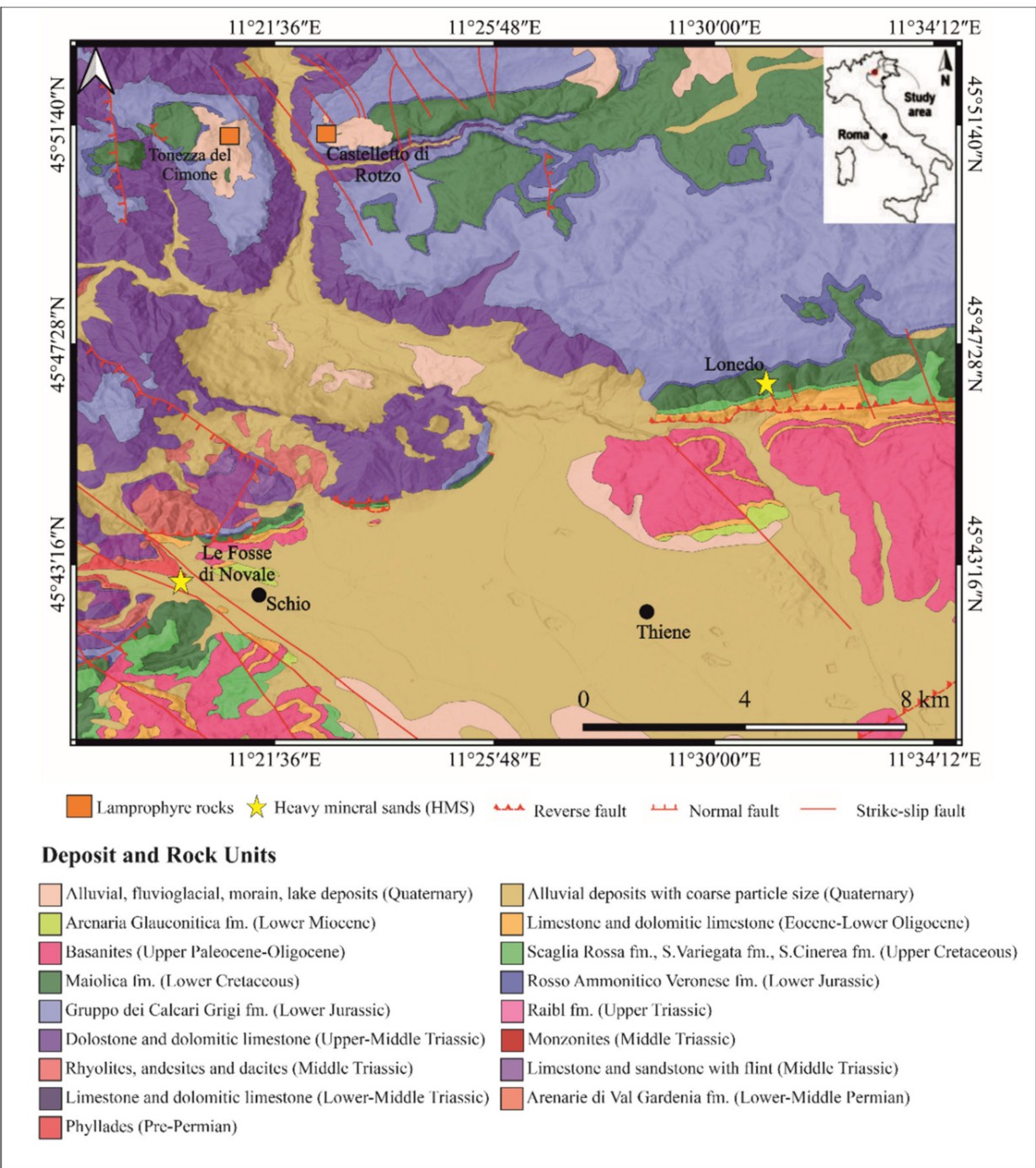

**Figure 2.** Geological map of the study area at Veneto compiled from literature [18] and new geological survey as part of this study.

## 3. Methods

The heavy minerals were separated using a sodium polytungstate aqueous solution (2.9 ρ), enabling rapid and effective mineral separations. After heavy-liquid separation, zircon crystals with sizes larger than 60 μm were hand-picked at the DiSPUTer, University "G. d'Annunzio" (Chieti, Italy). The smaller crystals, which are more affected by alteration due to the decay of U and Th, were not selected. Zircons have been studied by AX10 Zeiss optical microscope (Oberkochen, Germany) and Phenom XL SEM (Thermofisher Scientific, Waltham, MA, USA) housed at the geochemistry and volcanology laboratory, "G. d'Annunzio" University of Chieti-Pescara (Chieti, Italy).

All studies of multiphase and silicate melt inclusions in zircons from the Le Fosse di Novale soils were provided in the Institute of Geology and Mineralogy, Novosibirsk, Russia. Optical examination of the polished zircon grains mounted in epoxy resin (monitoring of multiphase inclusions) was performed using an Olympus BX51 microscope (Shinjuku, Tokyo, Japan). BSE images, elemental maps, and energy-dispersive spectroscopic (EDS) analyses of zircon and minerals from exposed inclusions were obtained using a MIRA 3LMU SEM (TESCAN Ltd., Brno, Czech Republic) equipped with an INCA Energy 450 XMax 80 microanalysis system (Oxford Instruments Ltd.). EDS analyses were performed using an accelerating voltage of 20 kV, a probe current of 1 nA, and an accumulation time of 20 s. The following simple compounds and metals were used as reference materials for most of the elements: $SiO_2$ (Si and O), $Al_2O_3$ (Al), diopside (Mg and Ca), albite (Na), orthoclase (K), $Ca_2P_2O_7$ (P), $BaF_2$ (Ba and F), $Cr_2O_3$ (Cr), pyrite (S), $CsRe_2Cl_6$ (Cl), metallic Ti, Fe, Mn, Zr, Hf and others. Correction for matrix effects was done using the XPP algorithm, implemented in the software of the microanalysis system. Metallic Co served for quantitative optimisation (normalisation to probe current and energy calibration of the spectrometer).

Electron Micro-Probe Analyser (EMPA) composition of zircons from soils of Novale was determined using the JEOL JXA 8200 Super-probe at Department of Earth Sciences, University of Milan (Milano, Italy), equipped with five WDS spectrometers, operating in wavelength dispersive mode. Operating conditions were 15 kV accelerating potential, 5 nA beam current, a spot size of 5 μm, and a counting time of 30 s on the peaks and 10 s on the backgrounds. The following natural minerals were used as reference materials: zircon Jarosevich for Zr, Hf and Si; ilmenite for Ti; Y-phosphate for Y; pure Nb.

Cathodoluminescence images were obtained using a Lumic HC6-LM cathodoluminescence microscope at the Natural History Museum, London. Trace element and U–Pb analysis was undertaken on zircons coming from soils and rocks, manually selected and picked up, embedded in epoxy resin before polishing. Analyses were performed using a 193 nm ESI Laser Ablation (LA) system coupled to an Agilent 7700 inductively coupled plasma mass spectrometer (ICP-MS). Laser acquisition was made with a spot size of 30 μm, frequency 5 Hz, fluence 3.5 J/cm$^2$, number of slices variable from 30 to 62 at the Natural History Museum, London. All the laser ablation data were reduced through an in-house Excel Spreadsheet. NIST SRM 610 glass and GEMOC zircon reference material GJ-1 were used as external reference materials for trace elements and U–Pb. Secondary reference materials NIST SRM 612, Plesovice and 91500 zircon reference materials were used to confirm analyses, and uncertainties fell within acceptable ranges. U–Pb measurements were obtained following analytical procedures proposed by [19].

Castelletto di Rotzo and Tonezza del Cimone rocks were studied by optical microscopy Zeiss and Phenom XL SEM, at the University "G. d'Annunzio" in Chieti (Chieti, Italy). Whole-rock geochemistry was performed at ActaLabs (Activation Laboratories, Ontario, Canada) by lithium metaborate/tetraborate fusion FUS-ICP for whole rock and FUS-MS for trace elements. FeO was detected by titration.

## 4. Results

### 4.1. Zircon Description

The Vicentine zircons are known as 'hyacinths' for the predominant reddish colour but often form complex shades ranging from brown to champagne (Figure 3). Zircon self-irradiance and lattice distortion confer colour that is of metamitic origin. For this reason, the zircon contained in recent rocks having an age of about 30–40 Ma is usually colourless, except for some primary colouration examples [20].

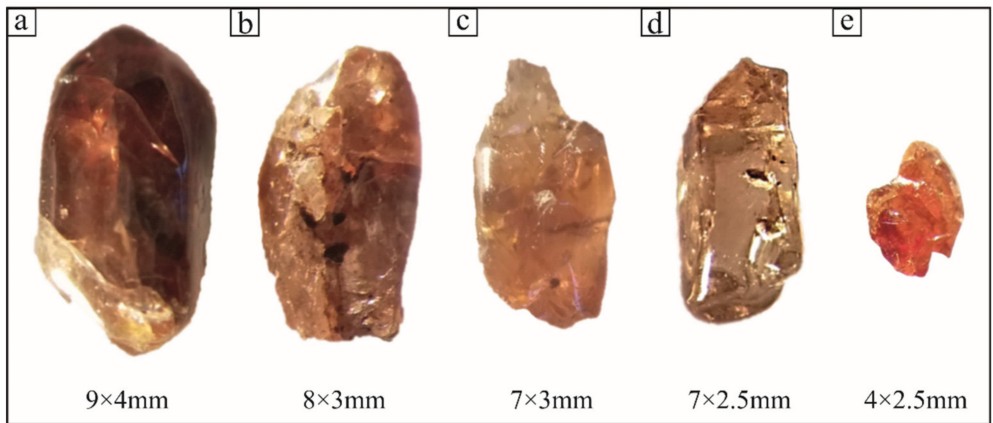

**Figure 3.** Representative crystals of zircon from Le Fosse di Novale (**a**–**c**,**e**) and Lonedo (**d**).

We analysed zircons from Le Fosse di Novale (13 grains: ZR-FN-1, ZR-FN-2, ZR-FN-3, ZR-FN-4, ZR-FN-5, VR1-2, VR1-3, VR1-4, VR2-1, VR2-2, VR2-3, VR3-1, VR3-4) and Lonedo (3 grains: ZR-LON 1, ZR-LON 2, ZR-LON 3) to obtain information on their provenance. In addition, zircons from the connate felsic-lithic from Castelletto di Rotzo lamprophyre were also analysed. The zircons are anhedral, subhedral and euhedral grains, sometimes with prismatic habit. They are mainly fractured, with poor/occasional cleavage and sometimes contain inclusions (Figure 4a–c). The zircons of Lonedo and Le Fosse di Novale are mainly homogenous in the BSE images (Figure 4d,f) and rarely show zonation (Figure 4e). The zonation possibly corresponds to the different chemical compositions in terms of average atomic weight. Other chemical differences are more clearly visible in cathodoluminescence images (Figure 4g–i). Figure 4g shows a zircon with pale and darker blue zonation and brownish shades, while Figure 4h,i shows yellow-brownish oscillatory zonation. Areas with a brighter CL response have a Th/U ratio of 0.4, while dark areas have a Th/U ratio of 0.86. Controversy about CL emissions' nature makes interpretation of zoned zircons not easy. Some authors have suggested that blue luminescent zircon contains more Hf than yellow luminescent zircon, richer in REE [21].

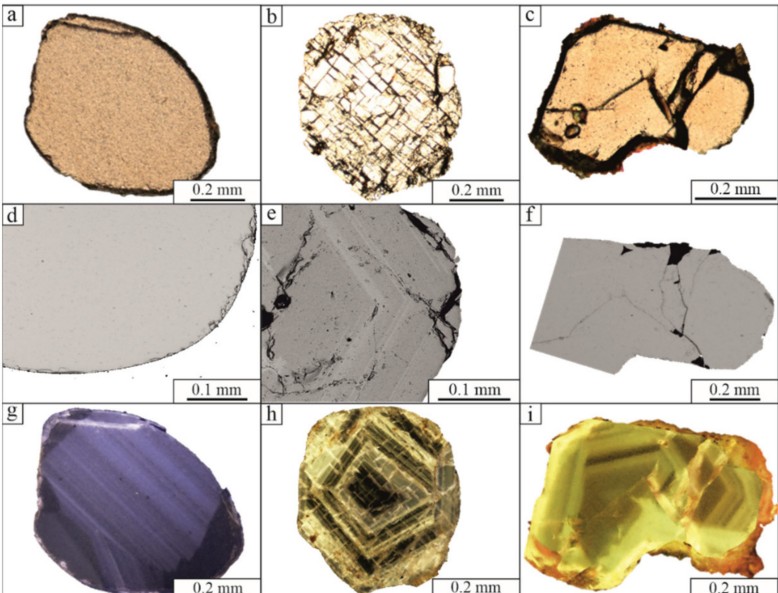

**Figure 4.** Image of three zircons from Lonedo (**a**,**d**,**g**); Le Fosse di Novale (**b**,**e**,**h**) and Castelletto di Rotzo (**c**,**f**,**i**). Optical image: (**a**–**c**); BSE images: (**d**,**e**,**f**); CL images: (**g**–**i**). Crystal b presents an unusual fracturing, maybe along cleavage planes. Crystal in figures (**e**,**g**–**i**) show an evident zonation.

### 4.2. Multiphase and Silicate-Melt Inclusions in the Le Fosse Di Novale Soils Zircons

Silicate-melt, fluid, mono- and poly-mineralic inclusions were observed in the Le Fosse di Novale zircons by optical microscopy and SEM-EDS (Figures 5–7). Inclusions are mainly confined to healed fractures in the host zircon. The silicate-melt and fluid inclusions together form isolated groups or trails in the grains (Figures 5 and 6). In general, the trails occur in the cores of the crystals and do not reach the zircon's outer zones, and their melt inclusions may be considered primary in origin. The sizes of silicate-melt inclusions are up to 10–15 μm. Most of these inclusions are glassy (silicate glass + one or two shrinkage bubbles), although rarely some of them are partially crystallised and contain daughter quenching phases in glass and calcite in the gas bubble (Table 1 and Figures 5 and 6). Silicate glass has trachy–phonolitic composition. It is rich in $Na_2O+K_2O$ (10.8–13.0 wt %) and $FeO_{total}$ (0.8–2.3 wt %) and poor in $TiO_2$ (0.2–0.4 wt %), CaO (0.3–1.1 wt %), MgO, $SO_3$ and Cl (up to 0.3 wt %) (Table 1).

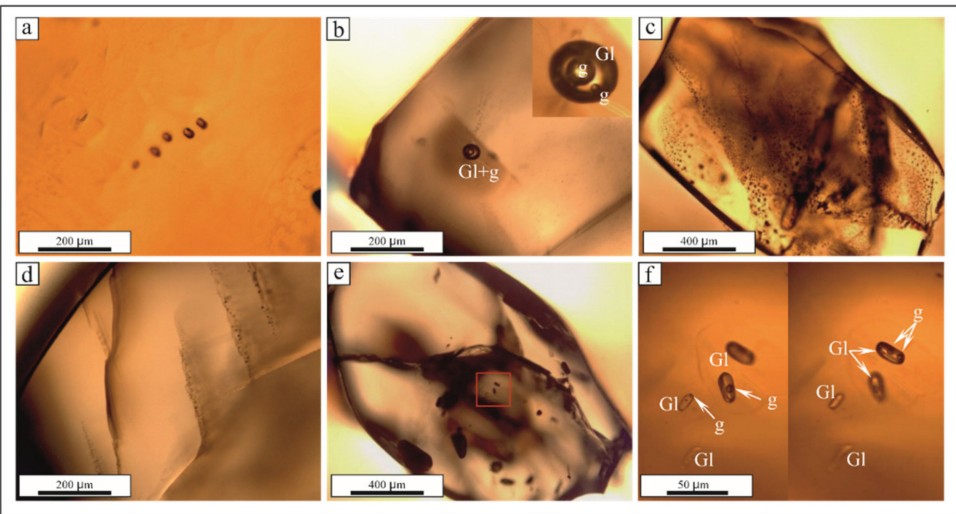

**Figure 5.** Groups and trails of silicate-melt and fluid inclusions in the zircons from Le Fosse di Novale transmitted light. (**a**) VR1-2; (**b**) VR1-4; (**c**) VR2-3; (**d**) VR3-6; (**e**) VR3-6; (**f**) VR3-5. Symbols: Gl—silicate glass, g—shrinkage gas bubble.

**Table 1.** Chemical composition (EDS, wt %) of glasses from silicate-melt inclusions in the zircons from Le Fosse di Novale.

| Inclusion | VR2-3-2 | VR2-3-3 | VR2-1-n2 | VR2-5-n2-1 | VR3-1-n1 | VR3-4-n1 | VR3-5-n2-1 | VR3-5-n2-2 |
|---|---|---|---|---|---|---|---|---|
| Phase Composition of Inclusion | Gl + g | Gl + g | Gl + Fe-oxi + Cc + g | Gl + Cc + g | Gl + g | Gl + g | Gl + g | Gl + g |
| *n* | *2* | *1* | *3* | *6* | *5* | *5* | *5* | *4* |
| $SiO_2$ | 58.11 | 54.87 | 55.73 | 57.80 | 55.44 | 56.84 | 56.26 | 57.09 |
| $TiO_2$ | 0.23 | 0.20 | 0.17 | 0.35 | 0.21 | 0.29 | 0.33 | 0.35 |
| $Al_2O_3$ | 17.17 | 17.27 | 21.06 | 18.89 | 18.59 | 18.08 | 17.79 | 18.19 |
| FeO | 1.80 | 1.81 | 0.81 | 2.11 | 1.41 | 2.21 | 2.19 | 2.25 |
| MnO | 0.00 | 0.00 | 0.00 | 0.13 | 0.16 | 0.00 | 0.13 | 0.13 |
| MgO | 0.20 | 0.15 | 0.00 | 0.27 | 0.13 | 0.23 | 0.23 | 0.21 |
| CaO | 0.82 | 0.80 | 0.34 | 1.09 | 0.38 | 0.97 | 1.00 | 0.92 |
| $Na_2O$ | 6.45 | 6.66 | 8.02 | 6.31 | 8.00 | 7.05 | 6.93 | 6.93 |
| $K_2O$ | 4.34 | 4.25 | 4.13 | 4.68 | 5.02 | 4.86 | 4.73 | 4.87 |
| $P_2O_5$ | 0.00 | 0.00 | 0.21 | 0.00 | 0.25 | 0.23 | 0.23 | 0.23 |
| $SO_3$ | 0.27 | 0.22 | 0.22 | 0.28 | 0.20 | 0.20 | 0.20 | 0.18 |
| Cl | 0.29 | 0.25 | 0.32 | 0.31 | 0.30 | 0.28 | 0.30 | 0.32 |
| Sum | 89.65 | 86.48 | 91.02 | 92.21 | 90.08 | 91.23 | 90.33 | 91.66 |

*N*—average. Symbols: Gl—silicate glass; g—shrinkage gas bubble; Fe-oxi–unidentified Fe-oxide; Cc—calcite. For details, see Figure 6b.

Mono- and poly-mineralic inclusions are commonly associated with silicate-melt and fluid ones (Figure 6). Hydroxylapatite, baddeleyite and calcite are the main components of such inclusions. Baddeleyite contains $HfO_2$ (0.9–1.5 wt %), $TiO_2$ (0.2–1.8 wt %) and FeO (0.3–0.8 wt %). Hydroxylapatite is rich in $Na_2O$ (0.7–1.0 wt %) and SrO (1.4–1.9 wt %), virtually free in F and contains $SiO_2$ (0.4–0.5 wt %), $SO_3$ (0.4–0.6 wt %) and Cl (0.5–0.7 wt %). Calcite sometimes contains MgO (up to 1.4 wt %). MnO and $FeO_{total}$ (up to 0.5 wt %). In addition, thorite ($Th_{1.04}(Si_{0.75}Sr_{0.04}P_{0.13}Ce_{0.04}O_4)$), yttrialite (($Y_{1.47}Th_{0.47})_{1.94}Si_{1.82}Ca_{0.09}Ce_{0.09}O_7$), aeschynite ($Y_{1.04}(Ti_{0.39}, Nb_{1.23}Si_{0.16}Th_{0.13})_{1.91}(O, OH)_6$), uraninite (($U_{0.71}Si_{0.08}Y_{0.18}Ce_{0.02})_{0.99}O_4$) have been identified in a polymineralic inclusion found in zircon from Lonedo (LON-2) (Figure 6h,i).

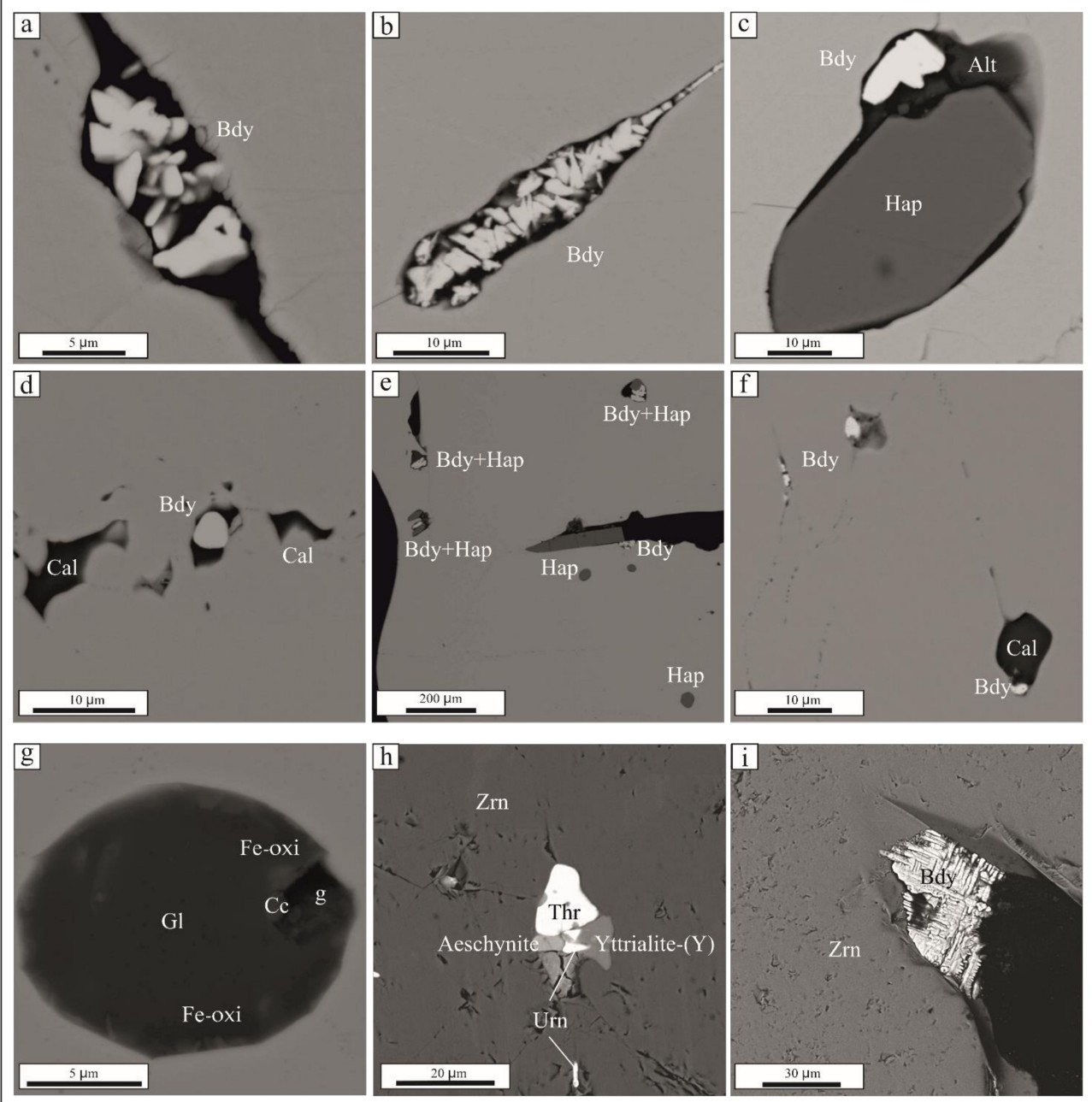

**Figure 6.** Mono- and poly-mineralic inclusions in the zircons from Le Fosse di Novale, BSE images. (**a**) VR1-3; (**b**) VR1-3-2; (**c**) VR2-1; (**d**) VR2-3-n1; (**e**) VR2-1-n1; (**f**) VR2-2-n2; (**g**) VR2-1; (**h**) aeschynite, yttrialite (Y), uraninite and thorite in zircon; (**i**) baddeleyite in zircon. Symbols: Bdy—baddeleyite, Hap—hydroxylapatite; Cal—calcite; Alt—alteration Al–Si-mineral (chlorite), Zrn—zircon, Thr—thorite, Urn—uraninite.

Salt compounds dominate some fluid inclusions. Figure 7 shows one of them that has been opened. It contains primarily sulphates and carbonates representing magmatic carbonatitic melt or salt-rich fluid. In addition, this inclusion contains glauberite $Na_2Ca(SO_4)_2$ and other alkali-rich Ca-sulphates, a $BaCa(CO_3)_2$ phase, baddeleyite, calcite, dolomite-siderite and minor Fe-sulphide and K-aluminosilicate. The presence of carbonates, gas bubbles ($CO_2$) and low closure may refer to the abundance of $CO_2$. Similar inclusion with trachytic glasses and carbonate salts have been found associated with carbonatitic rocks at San Venanzo, Italy [22].

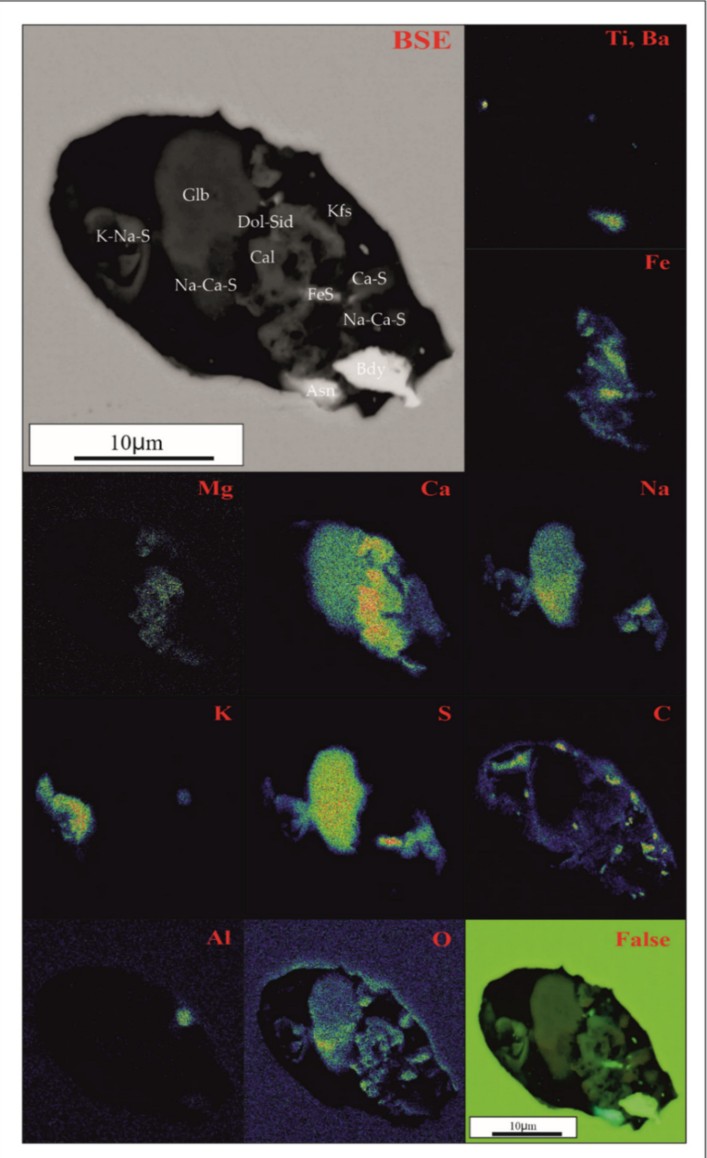

**Figure 7.** BSE image and elemental maps for a sulphate-carbonate inclusion in the Le Fosse di Novale zircon, grain VR2-2. Symbols: Bdy—baddeleyite; Asn—$BaCa(CO_3)_2$ mineral (alstonite, paraalstonite or barytocalcite); Glb—glauberite $Na_2Ca(SO_4)_2$; Dol-Sd—dolomite–siderite; Cal—calcite; K–Na–S, Na–Ca–S, Ca–S—unidentified K–Na, Na–Ca- and Ca-sulphates; FeS—Fe-sulphide; Kfs—potassic feldspar.

### 4.3. Zircon Geochemistry

The zircons of Castelletto di Rotzo, Le Fosse di Novale and Lonedo have semi-stoichiometric formula averaging $(Zr_{0.96}X_{0.04})_{1.00}Si_{1.00}O_4$, where X is for Hf, Y, Th and U contents. The contents of $HfO_2$, $Y_2O_3$, $ThO_2$ and $UO_2$ are up to 1.4 wt %, 0.3 wt %,

0.3 wt %, and 0.2 wt %, respectively. Other trace elements are up to 0.2 oxide wt %. The results of the analyses on the major elements (>0.01 oxide wt %) and trace elements (ppm)are given in Tables 2–5. The analytical condition and the complete analysis are provided as Supplementary Material.

**Table 2.** Representative EMPA analyses of the zircons from Le Fosse di Novale. $TiO_2$, $Nb_2O_5$ and $Y_2O_3$ are below detection limits (<0.01–0.05 wt %).

| Soils Zircons from Le Fosse di Novale | | | | | | | | | | | | | | | |
|---|---|---|---|---|---|---|---|---|---|---|---|---|---|---|---|
| **Sample** | **ZR-FN-2** | | | | | | | | | | | | | | **ZR-FN-4** |
| **Analysis** | #1 | #2 | #3 | #4 | #5 | #6 | #7 | #8 | #9 | #10 | #11 | #12 | #13 | #14 | #15 | #16 |
| $SiO_2$ | 32.8 | 32.8 | 32.5 | 32.8 | 32.8 | 32.8 | 32.8 | 32.9 | 32.8 | 32.9 | 32.8 | 32.8 | 32.8 | 32.9 | 32.9 | 32.8 |
| $ZrO_2$ | 65.9 | 66.3 | 66.2 | 66.9 | 66.7 | 66.8 | 66.6 | 66.9 | 67 | 67.1 | 67 | 66.9 | 67.1 | 67.2 | 67.2 | 66.7 |
| $HfO_2$ | 0.68 | 0.82 | 0.97 | 0.47 | 0.53 | 0.78 | 0.56 | 0.68 | 0.68 | 0.56 | 0.44 | 0.9 | 0.58 | 0.74 | 0.52 | 0.87 |

| **Sample** | **ZR-FN-4** | | | | | | | | | | | | | | |
|---|---|---|---|---|---|---|---|---|---|---|---|---|---|---|---|
| **Analysis** | #17 | #18 | #19 | #20 | #21 | #22 | #23 | #24 | #25 | #26 | #27 | #28 | #29 | #30 | Avg | σ |
| $SiO_2$ | 32.8 | 32.8 | 32.8 | 32.8 | 32.8 | 32.8 | 32.8 | 32.8 | 32.8 | 32.8 | 32.8 | 32.8 | 32.7 | 32.7 | 32.8 | 0.07 |
| $ZrO_2$ | 66.5 | 66.8 | 66.3 | 66.4 | 66.7 | 66.7 | 66.8 | 66.7 | 66.5 | 66.5 | 66.5 | 66.5 | 66.1 | 66.7 | 66.7 | 0.31 |
| $HfO_2$ | 1.08 | 0.39 | 1.01 | 0.99 | 0.51 | 0.71 | 0.89 | 0.64 | 0.63 | 0.67 | 0.69 | 1.02 | 1.24 | 0.93 | 0.74 | 0.21 |

**Table 3.** LA- ICP-MS zircon analyses (µg/g) from Castelletto di Rotzo.

| Area | Castelletto Di Rotzo | | | | | | | | | | | |
|---|---|---|---|---|---|---|---|---|---|---|---|---|
| **Sample** | **CASTELLETTO1** | | | | | | | | | | | |
| **Analysis** | #1 | #2 | #3 | #4 | #5 | #6 | #7 | #8 | #9 | #10 | #11 | #12 |
| Al | 1.2 | 0.72 | bdl | bdl | bdl | bdl | 0.74 | 0.77 | 0.94 | 8.6 | 0.71 | bdl |
| P | 193 | 193 | 79 | 125 | 124 | 88 | 84 | 64 | bdl | 82 | 150 | 123 |
| Ca | bdl | 190 | 160 | bdl | 160 | bdl | bdl | 170 | 190 | bdl | 160 | bdl |
| Ti | 3.37 | 3.35 | 3.84 | 4.21 | 5.6 | 4.89 | 4.14 | 2.92 | 2.9 | 3.19 | 5.1 | 2.27 |
| V | 0.09 | 0.09 | bdl | bdl | bdl | bdl | bdl | 0.09 | 0.11 | bdl | 0.09 | bdl |
| Fe | 20 | 20 | 21 | bdl | 20 | bdl | 20 | 22 | 25 | bdl | 20 | bdl |
| Y | 1094 | 1131 | 592 | 601 | 1081 | 819 | 806 | 358.3 | 240.3 | 572 | 1360 | 673 |
| Nb | 11 | 11.2 | 6.61 | 7.54 | 16.2 | 10.8 | 8.97 | 4.34 | 3.22 | 8.79 | 18.3 | 8.02 |
| Mo | 1.28 | 1.15 | 1.24 | 1.21 | 1.41 | 1.17 | 1.4 | 0.98 | 1.36 | 1.05 | 1.07 | 1.44 |
| Ba | 0.14 | bdl | bdl | bdl | 0.11 | 0.1 | bdl | bdl | 0.07 | bdl | bdl | bdl |
| La | 0 | 0.01 | bdl | bdl | 0.01 | 0 | 0.01 | bdl | 0.01 | 0.08 | 0.01 | 0.01 |
| Ce | 35.7 | 37.6 | 21.4 | 31.5 | 53.2 | 41.6 | 39.2 | 14.1 | 11.4 | 34.4 | 71.9 | 23.1 |
| Pr | 0.07 | 0.07 | 0.05 | 0.05 | 0.12 | 0.08 | 0.06 | 0.01 | bdl | 0.05 | 0.14 | 0.04 |
| Nd | 1.48 | 1.39 | 0.79 | 0.97 | 1.87 | 1.3 | 1.51 | 0.36 | 0.09 | 1.5 | 3.34 | 0.78 |
| Sm | 3.72 | 4.13 | 2.16 | 2.23 | 4.8 | 3.63 | 3.56 | 1.04 | 0.64 | 2.36 | 6.77 | 2.13 |
| Eu | 2.83 | 3 | 1.61 | 1.67 | 3.32 | 2.64 | 2.52 | 0.57 | 0.37 | 1.78 | 4.72 | 1.66 |
| Gd | 23.6 | 24.9 | 12.5 | 14.7 | 28.3 | 19.4 | 20 | 7.17 | 4.25 | 13.8 | 35.9 | 13.9 |
| Tb | 8.3 | 8.73 | 4.47 | 5.05 | 9.33 | 6.95 | 6.72 | 2.72 | 1.71 | 4.95 | 12.3 | 5.09 |
| Dy | 104 | 109 | 54.6 | 59.6 | 110 | 82.4 | 79.1 | 35.3 | 21.8 | 59.4 | 142 | 63 |
| Ho | 36.1 | 37.3 | 19.3 | 20.5 | 36.7 | 27.3 | 26.5 | 12.2 | 8.1 | 19.6 | 46.5 | 21.9 |
| Er | 156 | 163 | 84.7 | 86.1 | 152 | 115 | 112 | 53.3 | 34.2 | 80.7 | 194 | 97.1 |
| Tm | 30.3 | 30.6 | 16.2 | 15.8 | 27.7 | 21 | 20.9 | 9.78 | 6.48 | 14.8 | 33.8 | 18.7 |
| Yb | 239 | 244 | 132 | 125 | 216 | 166 | 163 | 77.3 | 51.8 | 118 | 265 | 152 |
| Lu | 42.3 | 43.2 | 24 | 21.7 | 36.3 | 28.3 | 27.6 | 13.9 | 9.31 | 19.6 | 44 | 27.3 |
| Hf | 8550 | 8340 | 8980 | 8970 | 8750 | 9070 | 8760 | 10750 | 10250 | 9460 | 9040 | 9200 |
| Ta | 2.81 | 2.89 | 2.66 | 3.27 | 5.04 | 3.87 | 3.4 | 2.18 | 1.69 | 3.41 | 5.94 | 2.43 |
| W | 0.02 | bdl | bdl | bdl | bdl | 0.06 | 0.04 | 0.08 | bdl | bdl | bdl | 0.06 |
| Pb | 4.41 | 4.65 | 2.27 | 3.95 | 4.88 | 3.8 | 3.45 | 1.05 | 0.98 | 3.53 | 5.66 | 2.61 |

bdl = below detection limit.

**Table 4.** LA-ICP-MS zircon analyses (μg/g) from Lonedo.

| Area | Lonedo | | | | | | | | | | | | | | | |
|---|---|---|---|---|---|---|---|---|---|---|---|---|---|---|---|---|
| Sample | Lon2 | | | | | | | | Lon3 | | | | | | | |
| Analysis | #1 | #2 | #3 | #4 | #5 | #6 | #7 | #8 | #1 | #2 | #3 | #4 | #5 | #6 | #7 | #8 |
| Al | 0.76 | bdl | 77.5 | 0.71 | bdl | 0.77 | 1.55 | 0.77 | 2.75 | 2.56 | 4.05 | 4.79 | 2.06 | 3.53 | 2.39 | 9.6 |
| P | 84 | 254 | 472 | 247 | 194 | 242 | 71 | 81 | 108 | bdl | 235 | 122 | 133 | 224 | 252 | 220 |
| Ca | bdl | bdl | 150 | bdl | bdl | 150 | bdl | 190 | 150 | 150 | 150 | bdl | 150 | bdl | bdl | 170 |
| Ti | 3.97 | 6.3 | 3.3 | 5.28 | 4.67 | 5.45 | 4.6 | 4.1 | 2.34 | 2.1 | 2.82 | 3.03 | 5.38 | 2.97 | 4.7 | 2.82 |
| V | bdl | 0.09 | 0.78 | bdl | 0.08 | bdl | 0.09 | 0.1 | 0.08 | 0.1 | 0.08 | 0.08 | 0.07 | 0.09 | bdl | 0.09 |
| Fe | bdl | bdl | 615 | bdl | 20 | bdl | 19 | bdl | bdl | 22 | bdl | 19 | 20 | bdl | bdl | bdl |
| Y | 425.9 | 1462 | 4360 | 1445 | 1157 | 1175 | 431.2 | 422.4 | 651 | 370 | 1246 | 795 | 1378 | 1154 | 1433 | 1258 |
| Nb | 4.87 | 13.2 | 177 | 12.9 | 9.12 | 10.2 | 4.95 | 5.11 | 8.28 | 7.21 | 16.2 | 11 | 51.3 | 14.7 | 19.7 | 16.8 |
| Mo | 1.22 | 0.83 | 1.02 | 0.99 | 1.22 | 1.12 | 0.99 | 0.99 | 1.09 | 1.31 | 0.95 | 1.06 | 1.22 | 1.07 | 1 | 1.01 |
| Ba | 0.16 | 0.11 | 2.74 | bdl | bdl | 0.11 | bdl | 0.14 | 0.14 | bdl | bdl | bdl | 0.2 | 0.1 | bdl | 0.11 |
| La | 0.01 | 0.01 | 0.48 | 0.01 | 0.01 | 0.01 | 0.01 | 0.01 | 0 | 0.01 | bdl | 0.01 | 0 | 0.01 | 0.01 | 0.02 |
| Ce | 19.3 | 59.4 | 433 | 53.2 | 42.4 | 45.4 | 19 | 18.8 | 10 | 8.3 | 15.7 | 10.3 | 39.5 | 15 | 17.5 | 15.9 |
| Pr | 0.03 | 0.14 | 1.78 | 0.15 | 0.14 | 0.13 | 0.04 | 0.03 | 0.02 | 0.02 | 0.07 | 0.04 | 0.16 | 0.06 | 0.07 | 0.08 |
| Nd | 0.73 | 2.66 | 26.5 | 2.79 | 2 | 2.23 | 0.59 | 0.74 | 0.46 | 0.67 | 0.94 | 0.8 | 2.95 | 0.9 | 1.22 | 1.29 |
| Sm | 1.41 | 6.79 | 42.7 | 7.07 | 5.79 | 5.1 | 1.84 | 1.65 | 1.43 | 0.69 | 3.72 | 2.39 | 6.05 | 3.1 | 4.33 | 3.47 |
| Eu | 1.05 | 4.37 | 22.2 | 4.3 | 3.81 | 3.47 | 1.11 | 1.08 | 1.35 | 0.67 | 2.29 | 1.46 | 4.09 | 2.13 | 2.57 | 2.27 |
| Gd | 9.16 | 37.4 | 173 | 36.2 | 29.8 | 28.5 | 10 | 9.06 | 11.4 | 7.2 | 22.7 | 15.2 | 36.1 | 20.6 | 25.8 | 22.8 |
| Tb | 3.49 | 12.5 | 50 | 12.5 | 10.1 | 9.74 | 3.53 | 3.29 | 4.86 | 2.84 | 9.52 | 5.93 | 13.1 | 8.86 | 10.7 | 9.59 |
| Dy | 39.8 | 148 | 507 | 145 | 115 | 118 | 40.3 | 40.3 | 61.7 | 36 | 126 | 79.8 | 156 | 115 | 140 | 125 |
| Ho | 13.5 | 47.5 | 149 | 47.3 | 38.1 | 38.9 | 13.8 | 13.6 | 22.1 | 12.4 | 42.9 | 27.4 | 48.1 | 39.1 | 49.4 | 43.9 |
| Er | 58.8 | 200 | 550 | 198 | 158 | 157 | 58.9 | 58.6 | 98.1 | 55.1 | 193 | 122 | 196 | 174 | 219 | 196 |
| Tm | 10.9 | 35.2 | 88.5 | 35.3 | 28.4 | 29.1 | 11.3 | 10.9 | 20.6 | 10.6 | 38 | 24 | 35.8 | 33.6 | 42.9 | 38 |
| Yb | 86.3 | 271 | 640 | 277 | 222 | 227 | 90 | 88.4 | 171 | 89 | 315 | 201 | 270 | 283 | 353 | 320 |
| Lu | 15.2 | 46.5 | 96.8 | 47 | 37.5 | 38.2 | 15.6 | 15.4 | 30.1 | 15.8 | 53.4 | 34.3 | 43 | 47.7 | 59.4 | 53.1 |
| Hf | 9680 | 8370 | 9660 | 8150 | 7740 | 8840 | 9360 | 9780 | 10810 | 11120 | 11940 | 12190 | 11970 | 11650 | 11970 | 12010 |
| Ta | 2.28 | 3.39 | 33.5 | 3.86 | 2.45 | 2.81 | 2.15 | 2.26 | 3.04 | 2.98 | 4.48 | 3.91 | 15 | 4.04 | 5.48 | 4.69 |
| W | bdl | 0.07 | 0.21 | 0.05 | 0.05 | bdl | bdl | bdl | 0.06 | 0.07 | 0.06 | bdl | bdl | 0.05 | bdl | 0.05 |
| Pb | 1.46 | 4.38 | 41.9 | 5.25 | 3.19 | 4 | 1.23 | 1.3 | 5.62 | 4.62 | 14.3 | 8.05 | 31 | 12.2 | 16.5 | 14.4 |

bdl = below detection limit.

**Table 5.** LA- ICP-MS zircon analyses (μg/g) from Le Fosse di Novale.

| Area | Novale | | | | | | | |
|---|---|---|---|---|---|---|---|---|
| Sample | ZR-FN-2 | | | | | | | |
| Analysis | #1 | #2 | #3 | #4 | #5 | #6 | #7 | #8 |
| Al | 13.8 | 12.6 | 2.49 | 6.03 | 4.01 | 3.27 | 8.7 | 48.8 |
| P | 99 | 276 | 162 | 197 | 108 | 214 | 272 | 250 |
| Ca | 160 | 170 | bdl | 160 | bdl | 180 | bdl | bdl |
| Ti | 2.66 | 16.1 | 7.7 | 10.8 | 4.2 | 7.8 | 19.3 | 17.7 |
| V | bdl | 0.08 | 0.1 | 0.09 | bdl | 0.08 | 0.08 | 0.11 |
| Fe | 21 | bdl | 19 | 19 | 20 | 20 | bdl | 19 |
| Y | 236 | 3369 | 2155 | 2194 | 758 | 2490 | 3273 | 3076 |
| Nb | 6.63 | 324 | 142 | 139 | 12.8 | 81.3 | 405 | 358 |
| Mo | 1.17 | 1.2 | 0.79 | 0.76 | 0.79 | 0.77 | 1.13 | 1.05 |
| Ba | bdl | 0.17 | bdl | 0.12 | 0.15 | bdl | 0.15 | 0.15 |
| La | 0.08 | 0.45 | 0.04 | 0.04 | bdl | 0.06 | 0.22 | 0.17 |
| Ce | 4.55 | 122 | 67.2 | 78.4 | 18 | 80.7 | 124 | 113 |
| Pr | 0.04 | 2.19 | 0.55 | 0.52 | 0.06 | 0.74 | 2.1 | 1.61 |
| Nd | 0.2 | 28.3 | 7.65 | 8.04 | 1.14 | 11 | 27.6 | 24.8 |
| Sm | 0.57 | 40.5 | 15.1 | 15.5 | 2.57 | 20.8 | 39.1 | 34.3 |
| Eu | 0.39 | 22.4 | 9.39 | 9.77 | 1.96 | 12.9 | 21.8 | 19.7 |
| Gd | 4.18 | 155 | 71.1 | 71.8 | 16.8 | 91.2 | 144 | 131 |

**Table 5.** *Cont.*

| Area | Novale | | | | | | | |
|---|---|---|---|---|---|---|---|---|
| Sample | ZR-FN-2 | | | | | | | |
| Analysis | #1 | #2 | #3 | #4 | #5 | #6 | #7 | #8 |
| Tb | 1.38 | 45.1 | 23.1 | 23.9 | 6.32 | 28.7 | 42.8 | 40 |
| Dy | 20.3 | 447 | 256 | 260 | 77 | 307 | 423 | 400 |
| Ho | 7.49 | 121 | 76 | 76.7 | 25.4 | 88.9 | 114 | 109 |
| Er | 39.4 | 441 | 300 | 295 | 111 | 339 | 419 | 401 |
| Tm | 8.54 | 74 | 51.6 | 50.6 | 20.7 | 58 | 70.1 | 67.2 |
| Yb | 79.3 | 532 | 396 | 377 | 168 | 426 | 510 | 491 |
| Lu | 15.6 | 79.8 | 61.6 | 58.3 | 28 | 65.1 | 76.7 | 73.7 |
| Hf | 13230 | 12910 | 12530 | 11980 | 10410 | 9700 | 12540 | 12420 |
| Ta | 5.74 | 77.5 | 31.8 | 26.7 | 4.88 | 16.5 | 92.7 | 79.8 |
| W | 0.05 | 0.12 | 0.05 | 0.03 | 0.06 | 0.04 | 0.03 | 0.05 |
| Pb | 5.26 | 84.7 | 47.4 | 44.8 | 7.59 | 32.2 | 92.9 | 83.9 |

bdl = below detection limit.

In the zircon crystal lattice, the chemical element Zr can be replaced by high ionic-charge $HFSE^{4+-5+}$ ($Ce^{4+}$, Hf, Ti, Th and U), while LILE or HFSE (ionic radius $> 10^{-10}$ m, such as La and $Ce^{3+}$) are incorporated depending on both charge and ionic radius, but non-tetravalent substitutions are coupled (e.g., $Zr^{4+} \leftrightarrow P^{5+}$ and $REE^{3+}$). LILE, HFSE and REE(chondrite normalised)against ionic potential are shown in Figure 8a,b. The zircons show a high U-Th, Zr-Hf, and Pb positive spike, as expected, and LILE depletion and a low Nb/Ta ratio. In general, zircon has high $HFSE^{4+}/HFSE^{5+}$ ratio. The Castelletto di Rotzo zircon is like the Lonedo zircons, while those from Fosse di Novale are much enriched in some HFSE, such as Th, U, Nb, and Ta. Castelletto di Rotzo zircons is depleted in HFSE but Zr-Hf. Vicentine zircons REE pattern show a sensible Ce positive spike and a very light Eu negative spike. Zircons from Castelletto di Rotzo have $\Sigma REE = 490$ µg/g and $La/Lu = 8 \times 10^{-4}$; Lonedo zircons have $\Sigma REE = 755$ µg/g and $La/Lu = 2 \times 10^{-3}$; Fosse di Novale zircons have $\Sigma REE = 1360$ µg/g and $La/Lu = 3 \times 10^{-3}$.

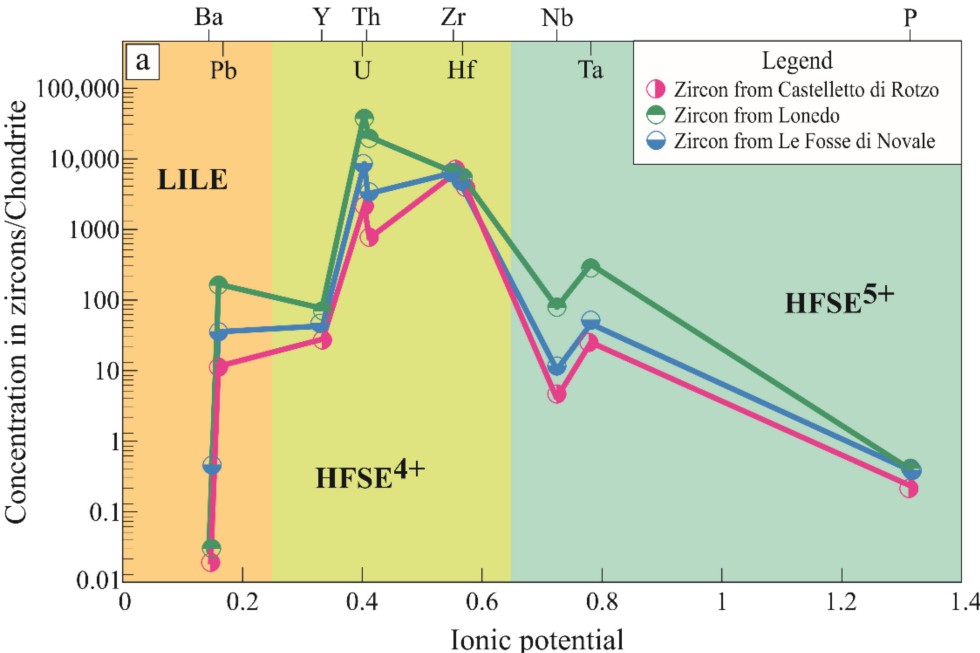

**Figure 8.** *Cont.*

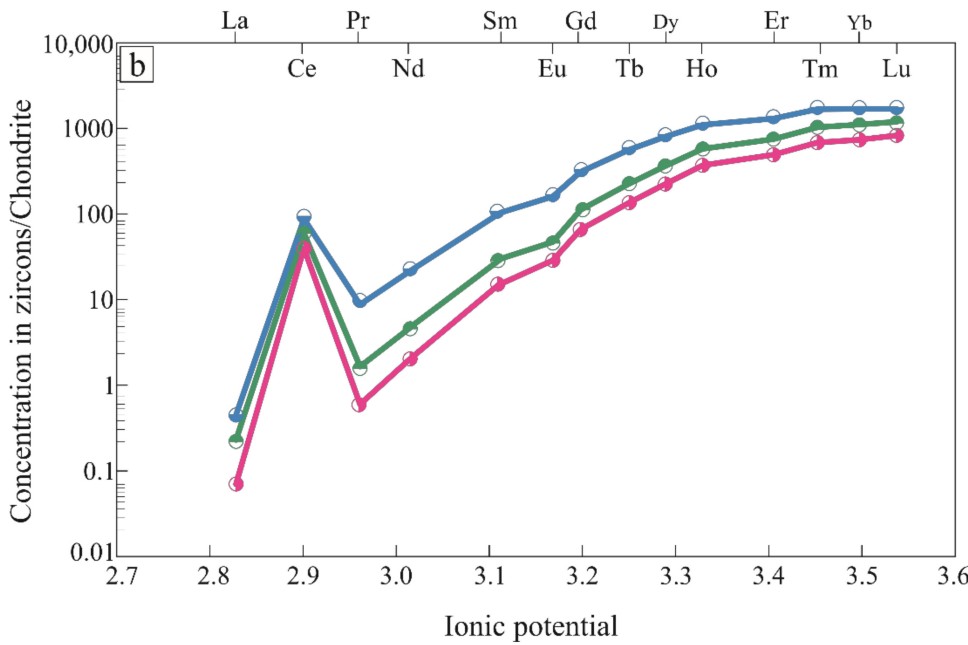

**Figure 8.** (**a**) Major Zr and trace element concentrations in zircon, normalised to chondrite [23], plotted as a function of ionic potential. (**b**) Zircon REE concentrations, normalised to chondrite [23], were plotted as a function of ionic potential.

### 4.4. Zircon Geochronology

Concordia ages and weighted mean $^{206}$Pb/$^{238}$U were calculated using the IsoplotR software [24]. The results are shown in Table 6, and the data were plotted on a conventional Wetherill Concordia diagram (Figure 9). The concordia age is 42.4 Ma for Castelletto di Rotzo, 44.6 Ma for Lonedo and 47.7 Ma for Le Fosse di Novale. Our data are different from those of Castelletto di Rotzo published by Visona et al. [4], where $^{206}$Pb/$^{238}$U ages vary between 29.4 and 31.6 Ma. The individual grain ages ($^{206}$Pb/$^{238}$U) vary between 45.7 to 38.6 Ma in this study. The ratio Th/U varies between 0.78 and 0.43. Uncertain ages reflect the analytical precision of the data collected. The pooled age precision would get better with more (consistent) analyses. We address this issue further statistically in the discussion section ahead. The ages for Lonedo and Le Fosse Di Novale agree with those of Visonà et al. [4], and thus, we are confident that they are representative.

### 4.5. Petrography

Alkaline and ultramafic lamprophyres, from Triassic to Lower Oligocene, including the Veneto ones, represent the majority of pre-Pleistocene alkaline mafic magmatic activity in Italy [25,26]. For the classification of the lamprophyres, we use the scheme proposed by Streckeisen [27]. There is no distinction between phenocrysts and groundmass in this scheme, but it bases on the relationships between felsic minerals (potassic feldspar vs. plagioclase, feldspar vs. feldspathoids) and the paragenesis of mafic minerals (Table 7).

The zircon host-rock was unknown so far for the studied samples. Therefore, we restrict the provenance hypothesis based on zircon age (Table 6), composition, and the feeding, glacial-river basin geology. Visonà et al. [4] proposed the lamprophyric rock of Castelletto di Rotzo as a zircon host-rock. Our investigation extended to the nearby Tonezza dikes and diatremes swarm located 2.5 km E of Castelletto di Rotzo where thermal metamorphism is extensive.

**Table 6.** Results of U–Pb age calculation. Rho: error correlation of $^{207}Pb/^{235}U$ and $^{206}Pb/^{238}U$ defined as $(err^{206}Pb/^{238}U)/(err^{207}Pb/^{235}U)$.

| Sample # | $Pb^{207}/U^{235}$ | 2s | $Pb^{206}/U^{238}$ | 2s | Rho | Date_$^{207}Pb/^{235}U$ (Ma) | 2s | Date_$^{206}Pb/^{238}U$ (Ma) | 2s | U (ppm) | Th (ppm) | Th/U |
|---|---|---|---|---|---|---|---|---|---|---|---|---|
| Castelletto 1 | 0.045 | 0.0160 | 0.007 | 0.0003 | 0.1898 | 43.0 | 15.0 | 43.0 | 2.1 | 169 | 107 | 0.63 |
| Castelletto 2 | 0.042 | 0.0140 | 0.007 | 0.0004 | 0.1169 | 41.0 | 13.0 | 43.8 | 2.5 | 175 | 113 | 0.65 |
| Castelletto 3 | 0.059 | 0.0330 | 0.007 | 0.0006 | 0.0810 | 52.0 | 31.0 | 44.3 | 3.8 | 84 | 44 | 0.52 |
| Castelletto 4 | 0.047 | 0.0190 | 0.007 | 0.0004 | 0.0260 | 44.0 | 18.0 | 45.7 | 2.4 | 141 | 91 | 0.65 |
| Castelletto 5 | 0.043 | 0.0160 | 0.007 | 0.0004 | 0.0790 | 41.0 | 15.0 | 43.4 | 2.5 | 183 | 142 | 0.78 |
| Castelletto 6 | 0.044 | 0.0150 | 0.006 | 0.0004 | 0.1101 | 42.0 | 14.0 | 39.9 | 2.3 | 154 | 109 | 0.71 |
| Castelletto 7 | 0.041 | 0.0180 | 0.007 | 0.0004 | 0.1021 | 43.0 | 19.0 | 42.8 | 2.6 | 131 | 87 | 0.66 |
| Castelletto 8 | 0.007 | 0.0710 | 0.006 | 0.0008 | 0.0958 | -12.0 | 65.0 | 38.6 | 4.9 | 46 | 21 | 0.46 |
| Castelletto 9 | 0.104 | 0.0830 | 0.007 | 0.0012 | 0.2303 | 79.0 | 76.0 | 43.2 | 7.5 | 36 | 15 | 0.43 |
| Castelletto 10 | 0.035 | 0.0160 | 0.006 | 0.0005 | 0.1517 | 33.0 | 16.0 | 40.9 | 3.1 | 143 | 117 | 0.82 |
| Castelletto 11 | 0.031 | 0.0120 | 0.006 | 0.0003 | 0.0573 | 33.0 | 13.0 | 40.8 | 1.9 | 228 | 180 | 0.79 |
| Castelletto 12 | 0.041 | 0.0240 | 0.007 | 0.0005 | 0.1520 | 37.0 | 23.0 | 41.8 | 3.1 | 101 | 49 | 0.48 |
| Lon2_1 | 0.011 | 0.0370 | 0.007 | 0.0007 | 0.1466 | 2.0 | 36.0 | 43.4 | 4.7 | 54 | 29 | 0.54 |
| Lon2_2 | 0.031 | 0.0170 | 0.007 | 0.0003 | 0.0370 | 29.0 | 16.0 | 43.0 | 2.1 | 166 | 146 | 0.88 |
| Lon2_3 | 0.042 | 0.0048 | 0.007 | 0.0002 | 0.0126 | 41.8 | 4.7 | 42.2 | 1.2 | 1647 | 3130 | 1.90 |
| Lon2_4 | 0.041 | 0.0120 | 0.007 | 0.0004 | 0.2087 | 40.0 | 12.0 | 42.6 | 2.4 | 203 | 181 | 0.89 |
| Lon2_5 | 0.039 | 0.0170 | 0.007 | 0.0005 | 0.0956 | 37.0 | 16.0 | 43.0 | 2.9 | 121 | 95 | 0.78 |
| Lon2_6 | 0.031 | 0.0150 | 0.007 | 0.0004 | 0.0430 | 30.0 | 15.0 | 43.6 | 2.6 | 150 | 113 | 0.76 |
| Lon2_7 | 0.034 | 0.0430 | 0.006 | 0.0006 | 0.1220 | 22.0 | 41.0 | 39.7 | 4.0 | 51 | 27 | 0.54 |
| Lon2_8 | 0.023 | 0.0310 | 0.006 | 0.0007 | 0.1918 | 16.0 | 31.0 | 40.9 | 4.7 | 52 | 28 | 0.54 |
| Lon3_1 | 0.047 | 0.0150 | 0.007 | 0.0003 | 0.0055 | 46.0 | 14.0 | 44.6 | 2.2 | 206 | 99 | 0.48 |
| Lon3_2 | 0.062 | 0.0210 | 0.007 | 0.0006 | 0.0101 | 60.0 | 20.0 | 45.4 | 3.9 | 165 | 80 | 0.49 |
| Lon3_3 | 0.047 | 0.0062 | 0.007 | 0.0002 | 0.0781 | 46.5 | 6.0 | 46.5 | 1.5 | 503 | 329 | 0.65 |
| Lon3_4 | 0.042 | 0.0100 | 0.007 | 0.0003 | 0.1901 | 41.5 | 9.7 | 45.2 | 1.9 | 291 | 157 | 0.54 |
| Lon3_5 | 0.045 | 0.0059 | 0.007 | 0.0002 | 0.1596 | 44.8 | 5.7 | 46.0 | 1.2 | 1101 | 1089 | 0.99 |
| Lon3_6 | 0.050 | 0.0073 | 0.007 | 0.0003 | 0.2100 | 49.5 | 7.0 | 44.2 | 1.6 | 450 | 273 | 0.61 |
| Lon3_7 | 0.050 | 0.0063 | 0.007 | 0.0002 | 0.1545 | 49.1 | 6.1 | 45.8 | 1.4 | 588 | 406 | 0.69 |
| Lon3_8 | 0.048 | 0.0063 | 0.007 | 0.0003 | 0.1602 | 47.2 | 6.0 | 45.9 | 1.6 | 510 | 329 | 0.65 |
| Novale2 | 0.054 | 0.0180 | 0.007 | 0.0004 | 0.1371 | 52.0 | 17.0 | 47.8 | 2.7 | 186 | 81 | 0.44 |
| Novale2_1 | 0.050 | 0.0033 | 0.007 | 0.0002 | 0.1234 | 49.0 | 3.2 | 47.7 | 1.0 | 2905 | 4830 | 1.66 |
| Novale2_2 | 0.044 | 0.0035 | 0.007 | 0.0001 | 0.1548 | 43.9 | 3.4 | 46.5 | 0.9 | 1673 | 2310 | 1.38 |
| Novale2_3 | 0.051 | 0.0036 | 0.007 | 0.0002 | 0.0085 | 50.5 | 3.4 | 47.1 | 1.1 | 1556 | 2393 | 1.54 |
| Novale2_4 | 0.044 | 0.0093 | 0.007 | 0.0004 | 0.1293 | 43.3 | 9.0 | 47.8 | 2.3 | 257 | 156 | 0.61 |
| Novale2_5 | 0.052 | 0.0041 | 0.007 | 0.0002 | 0.1904 | 51.3 | 4.0 | 48.1 | 1.1 | 1088 | 1604 | 1.47 |
| Novale2_6 | 0.050 | 0.0030 | 0.008 | 0.0001 | 0.3744 | 49.1 | 2.9 | 48.7 | 0.8 | 3131 | 5070 | 1.62 |
| Novale2_7 | 0.046 | 0.0030 | 0.007 | 0.0002 | 0.2736 | 45.2 | 2.9 | 47.7 | 1.0 | 2881 | 4600 | 1.60 |

The first description of the Castelletto di Rotzo rocks is by Ogniben [28] and De Vecchi [17], who focused on basic mineralogy. More mineralogical data were published by Boscardin et al. [29,30]. De Vecchi [17] classified the rock as a porphyritic camptonite. Zorzi et al. [5] published a list of minerals from the Castelletto di Rotzo rocks, including many accessory minerals. Castelletto di Rotzo shows two main different subvolcanic occurrences and facies, massive (dyke) (Figure 10a,b) and brecciated facies (diatreme) (Figure 10c,d). The rock contains abundant mantle (spinel lherzolite), felsic xenoliths (calcite–syenite), and felsic megacrysts. The dyke rock is fine-grained with ocellar facies at the contact with the country-rock. It is porphyritic with olivine crystals ($Fo_{91}$) (partially altered to lizardite), anhedral cpx (diopside), K-feldspar (sanidine and anorthoclase), in an interstitial groundmass composed of sub-parallel arranged elongated prisms of skeletal cpx (augite-salite), mica flakes (phlogopite), amphibole (Ca–amphibole–kaersutite) and glass.

Accessory minerals are ulvospinel, magnetite, pyrite, apatite, calcite. It is an ultramafic rock with less than 10% of felsic minerals. Ocelli are mainly mosaic textured calcite with a chlorite-group mineral reaction rim and coalescent with menisci necks. The rock texture suggests rapid quench and ocelli migration towards the wall-rock. Away from the contact, the rock becomes holocrystalline inequigranular, medium-grained with discrete megacrysts of olivine and K-feldspar, in a microcrystalline, intergranular groundmass of cpx (diopside-augite), rare haüyne and nepheline, mica (phlogopite and biotite) and amphibole (kaersutite). Accessory minerals are ulvospinel, magnetite, apatite, and calcite. Average modal composition is 32% clinopyroxene, 27% olivine, 11% carbonates, 7% K-feldspar, 5% orthopyroxene, 6% apatite, 4% mica, 3% oxides, 2% amphibole, 1% pyrite and 2% other unidentified phases. These rocks are classed as ultramafic, having felsic minerals <10 vol. %.

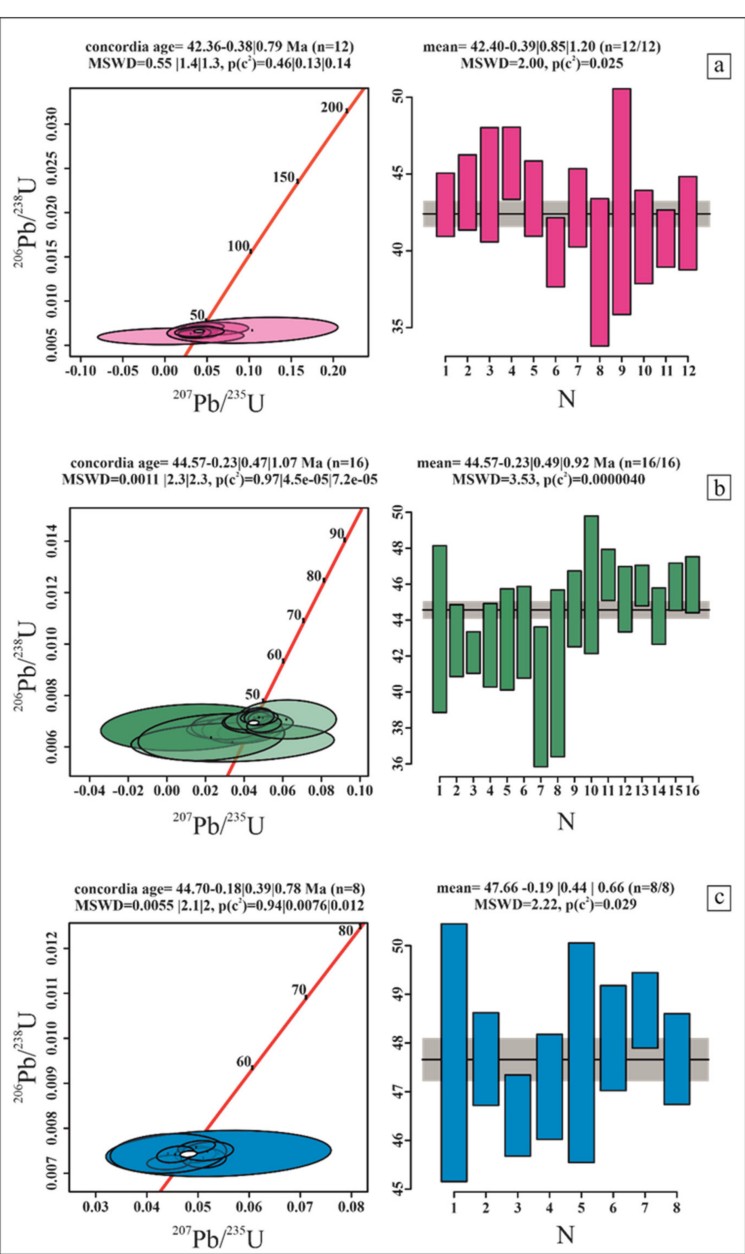

**Figure 9.** Concordia diagram and weighted mean $^{206}Pb/^{238}U$ age (Ma) for samples (**a**) Castelletto di Rotzo, (**b**) Lonedo, and (**c**) Le Fosse di Novale. The grey lines in the weighted mean plots show the calculated pooled ages for each sample.

**Table 7.** Lamprophyre classification (Streckeisen, 1978) [27].

| Light-Coloured Constituents | | Predominant Mafic Minerals | | | |
|---|---|---|---|---|---|
| Feldspar | Feldespathoid | Biotite, diopside, augite ($\pm$olivine) | Horneblende, diopside, augite ($\pm$olivine) | Alkaline amphibole, titanoaugite, olivine, biotite | Melilite, biotite, $\pm$titanoaugite, $\pm$olivine, $\pm$calcite |
| Or>pl | - | Minette | Vogesite | - | - |
| Pl>or | - | Kersantite | Spessartite | - | - |
| Or > pl | Felds > foid | - | - | Sannaite | - |
| Pl > or | Felds > foid | - | - | Camptonite | - |
| - | Glass or foid | - | - | Monchiquite | Polzenite |
| - | - | - | - | - | Alnoite |

The breccia conduit (diatreme) contains abundant mantle debris and mantle xenoliths (spinel clinopyroxenite-lherzolite, Figure 10e,f) plus felsic holocrystalline xenoliths (calcite–syenite), K-feldspar megacrysts and limestones fragments with a thin sparitic calcite reaction rim. The dyke clasts (autoclasts from the diatreme) have a plastic shape resembling 'ignimbrite fiammae'. They represent high-temperature subvolcanic lapilli produced by breccia fluidisation operated by $CO_2$–$H_2O$-rich fluids. Subvolcanic tuff represents a pyroclastic variant named tuffisite [5,31]. The breccia matrix comprises the same minerals found in the dyke-rock plus miaroles filled by small laths of K-feldspar, orthopyroxene, amphibole, calcite, baryte, hydrogarnet (hydrogrossular), olivine ($Fo_{85}$), ulvospinel, magnetite, and pyrite. The breccia is pervasively veined by apophyllite, calcium-silicate-hydrates (CSH), i.e., tobermorite-group minerals; calcium-aluminum-silicate-hydrates (CASH); i.e., thaumasite, and some Ca-K-zeolites, i.e., thomsonite and phillipsite. This association is typical of contact rocks from alkaline intrusions, by reaction with sedimentary or magmatic carbonates and deposition by hydrothermal fluids [32]. Average modal composition of the breccia is 34% clinopyroxene, 29% olivine, 9% orthopyroxene, 8% K-feldspar, 8% carbonates, 3% oxides, 3% apatite and 3% amphiboles. Feldspathoids, pyrite, zircon, REE-phases, CSHs and CASHs are less than 1%. The modal composition is modified by the disaggregation of minerals from xenoliths both mantle and felsic. The rock is ultramafic, having felsic minerals <10 vol. %.

The variations in the rock texture result from crystallisation of the same magma occurring under two different conditions: rapid cooling at contact with the wall-rock and slower cooling away from the contact. Rapid, volatile-rich magma flow at the dyke core produced a conduit filled with a breccia carrying mantle and crustal xenoliths. Modally, hydrated minerals and the presence of feldspar in the groundmass and the mode of occurrence of these rock-facies allow for their classification using the scheme of lamprophyric rocks (IUGS, [33]). The presence of foids and glass in the groundmass in low quantities, the abundance of K-feldspar and the presence of orthoclase in greater quantity than (virtually absent) plagioclase suggests that the rock of Castelletto di Rotzo classifies as sannaite and not camptonite, as suggested by De Vecchi [17].

Like the host sannaite lamprophyres, felsic xenoliths may be the host rock for zircon. They are fresh, inequigranular and mainly composed of sizeable perthitic anorthoclase and calcite crystals sometimes intergrowth with K-feldspar (Figure 10g,h). In the syenites, calcite forms amoeboid, interstitial monocrystals in apparent textural equilibrium with other silicate and phosphate minerals. The carbonate is predominantly calcic, with trace amounts of Sr. There are rare large apatite and zircon crystals. K-feldspar contains rare inclusions of biotite, REE silicates, phosphates, and silicate-phosphates (chevkinite ($Ce_{1.75}La_{1.24}Ca_{0.75}$)$_{3.74}$($Fe_{1.24}Mg_{0.18}Al_{0.99}$)$_{2.41}$($Ti_{2.32}Al_{0.68}$)$_3Si_4O_{22}$), monazite and an unidentified REE-phosphate-silicates like britholite. Most of the megacrysts of anorthoclase found in the rock come from these xenoliths. Megacrysts can be rounded with a tiny reaction rim of CSH or sharp angular fragment with no signs of reaction.

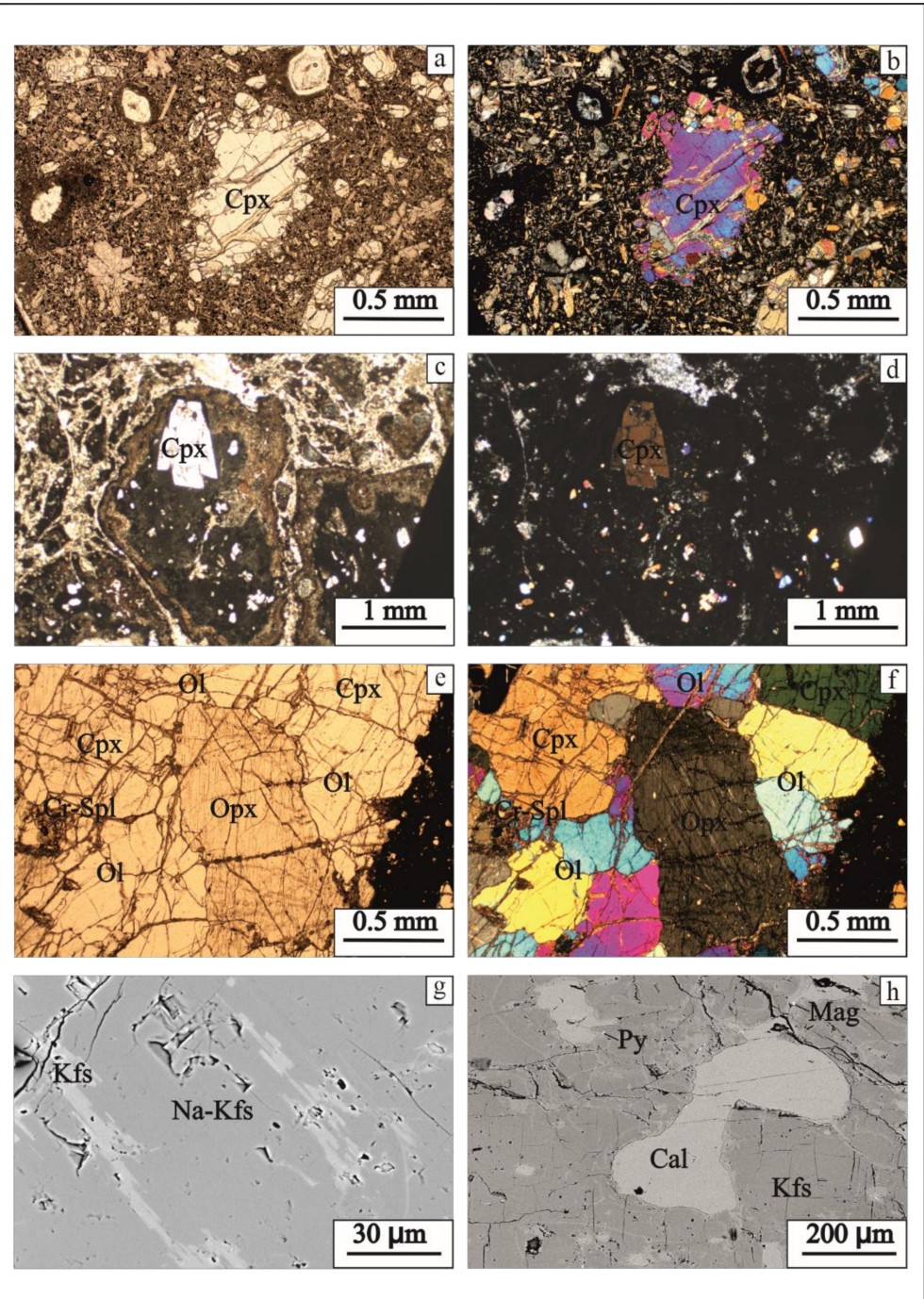

**Figure 10.** Image optical microscope. (**a**,**b**) Massive facies with ocelli. In the centre, corroded cpx phenocrysts (Augite) in plane-polarised light augite) (// and X- (**a**), which shows high birefringence colours under crossed polarised light; respectively); (**c**,**d**) brecciated facies shows a series of parallel and crossed polarised light showing auto-lapilli (// and X-Nicols); (**e**,**f**) plane and crossed polarised light images of nodules including a lherzolite nodule from the brecciated facies showing granular clinopyroxene, olivine, orthopyroxene with clinopyroxene exsolution lamellae exsolutions, and accessory Cr–spinel and amphibole. BSE Images. (**g**) Calcite–syenite xenolith showing K-feldspar exsolution perthites in Na-feldspar (BSE image); (**h**) calcite–syenite showing intergranular phenocrysts primary calcite, pyrite, and magnetite inclusion in K-feldspar. Symbols: Cpx—clinopyroxene; Ol—olivine; Opx—ortopyroxene; Kfs—k-feldspar; Py—pyrite; Mag—magnetite; Cal—calcite.

The Tonezza del Cimone dykes are olivine foidite with ameboid ocelli and textures indicating a degree of plastic behaviour. The rock is equigranular, fine-grained and presents a fluidal intergranular texture due to the isorientated clinopyroxene crystals. The rock is composed of slightly altered olivine and clinopyroxene; there are abundant nepheline, apatite, and clinopyroxene in the groundmass Ti-magnetite and rare phlogopite and amphibole. Large mantle xenoliths (several cm) are lherzolite. The provisional rock classification is monchiquite.

### 4.6. Whole-Rock Geochemistry

The Castelletto di Rotzo rock is ultramafic, with $SiO_2$ varying between 43–37.8 wt %; metaluminous $Na_2O + K_2O$ (1.26) < $Al_2O_3$(10.9) < $Na_2O + K_2O + CaO$ (21) with Agpaitic Index (AI) averaging 0.28 and is potassic $K_2O > Na_2O$-2. The Mg# ($Mg/Mg + Fe^{2+}$) is 0.77 and Cr + Ni average 540, indicating a very primitive mantle origin of the parental melt, corroborated by mantle debris and nodules (Table 8).

**Table 8.** Whole-rock geochemical data table (major element in wt %; trace elements and REE in µg/g)).

| Area | | Castelletto Di Rotzo | | Tonezza del Cimone | Area | | Castelletto Di Rotzo | | Tonezza del Cimone |
|---|---|---|---|---|---|---|---|---|---|
| **Major Elements** | | | | | **Trace Elements** | | | | |
| Sample | Detection Limit | VEN19-01 | VEN19-02 | VEN18-01 | Sample | Detection Limit | VEN19-01 | VEN19-02 | VEN18-01 |
| $SiO_2$ | 0.01 | 40.1 | 37.8 | 42.9 | Be | 1 | 2 | 3 | 2 |
| $TiO_2$ | 0.001 | 2.73 | 2.64 | 2.54 | Sc | 1 | 21 | 20 | 16 |
| $Al_2O_3$ | 0.01 | 10.5 | 11.4 | 14.6 | V | 5 | 266 | 244 | 193 |
| $Fe_2O_3$ | 0.01 | 5.07 | 7.87 | 5.06 | Cr | 20 | 400 | 200 | 210 |
| FeO | 0.1 | 6.9 | 4 | 5.7 | Co | 1 | 59 | 44 | 38 |
| MnO | 0.001 | 0.18 | 0.17 | 0.12 | Ni | 20 | 330 | 150 | 150 |
| MgO | 0.01 | 12.7 | 7.4 | 9.92 | Cu | 10 | 60 | 40 | 40 |
| CaO | 0.01 | 12 | 13.1 | 5.66 | Zn | 30 | 120 | 110 | 130 |
| $Na_2O$ | 0.01 | 1.83 | 0.43 | 2.14 | Ga | 1 | 19 | 18 | 20 |
| $K_2O$ | 0.01 | 1.72 | 2.15 | 2.58 | Ge | 1 | 1 | 1 | <1.00 |
| $P_2O_5$ | 0.01 | 1.14 | 1.28 | 1.28 | As | 5 | <5.00 | <5.00 | <5.00 |
| LOI | | 4.8 | 9.89 | 6.11 | Rb | 2 | 42 | 46 | 49 |
| Total | | 100 | 98.3 | 99.6 | Sr | 2 | 1057 | 794 | 962 |
| Mg# | | 76.6 | 76.7 | 75.6 | Y | 1 | 22 | 26 | 23 |
| AI. | | 0.33 | 0.23 | 0.32 | Zr | 2 | 236 | 239 | 311 |
| **Rare Earth Elements** | | | | | Nb | 1 | 83 | 98 | 74 |
| Sample | Detection Limit | VEN19-01 | VEN19-02 | VEN18-01 | Mo | 2 | 4 | <2.00 | <2.00 |
| La | 0.1 | 62.2 | 67.4 | 67.6 | Ag | 0.5 | 0.7 | 0.8 | 0.8 |
| Ce | 0.1 | 119 | 125 | 125 | In | 0.2 | <0.20 | <0.20 | <0.20 |
| Pr | 0.05 | 13.5 | 14.3 | 13.6 | Sn | 1 | 2 | 2 | 2 |
| Nd | 0.1 | 54 | 56.2 | 51 | Sb | 0.5 | <0.50 | <0.50 | <0.50 |
| Sm | 0.1 | 10.6 | 10.5 | 9.4 | Cs | 0.5 | 0.7 | 0.5 | <0.50 |
| Eu | 0.05 | 3.15 | 3.42 | 2.99 | Ba | 2 | 652 | 780 | 735 |
| Gd | 0.1 | 8.7 | 9.7 | 7.5 | Hf | 0.2 | 5.2 | 5.6 | 5.6 |
| Tb | 0.1 | 1.2 | 1.3 | 1.1 | Ta | 0.1 | 4.6 | 5.3 | 4.8 |
| Dy | 0.1 | 5.7 | 6.2 | 5.8 | W | 1 | 1 | 1 | <1.00 |
| Ho | 0.1 | 0.9 | 1.1 | 1 | Tl | 0.1 | 0.2 | <0.10 | <0.10 |
| Er | 0.1 | 2.5 | 2.8 | 2.5 | Pb | 5 | <5.00 | <5.00 | <5.00 |
| Tm | 0.05 | 0.31 | 0.36 | 0.32 | Bi | 0.4 | <0.40 | <0.40 | <0.40 |
| Yb | 0.1 | 1.7 | 2.1 | 1.9 | Th | 0.1 | 7.6 | 8.7 | 7.6 |
| Lu | 0.01 | 0.24 | 0.26 | 0.27 | U | 0.1 | 2.1 | 2.4 | 2.2 |

In the conventional TAS rock-classification diagrams, the Castelletto di Rotzo rocks plot in the foidite field and Tonezza in the tephrite-basanite field. However, the TAS diagram should be not used for rocks having a LOI > 2, so composition is merely indicative. Their compositions are different from glassy inclusions in the zircons, which are relatively homogeneous, trachy-phonolitic in composition and may correspond to the calcite–syenite connate lithic composition (Table 1). The R1-R2 diagram can also be used for rocks having

LOI>2 but require $CO_2$ estimation, which is impossible using SEM-EDS analyses of inclusions. However, using the R1-R2 diagram, the glasses plot in the same rock-type field of TAS.

The primitive mantle normalised multielement diagram (Figure 11a) shows a smooth semi-bell-shaped curve due to a relatively low LILE/HFSE$^{4+5+}$ fractionation and strong fractionation of LREE/HREE. A slight negative anomaly of K, Sr and Y are apparent. REE$_N$ patterns (Figure 11b) show a (La/Lu)$_N$ is about 31 for all the rocks. There is no Eu anomaly, usually seen in rocks with plagioclase. The Castelletto di Rotzo and Tonezza del Cimone lamprophyres have a similar element distribution, thus indicating a common source and partial melting degree (Figure 11).

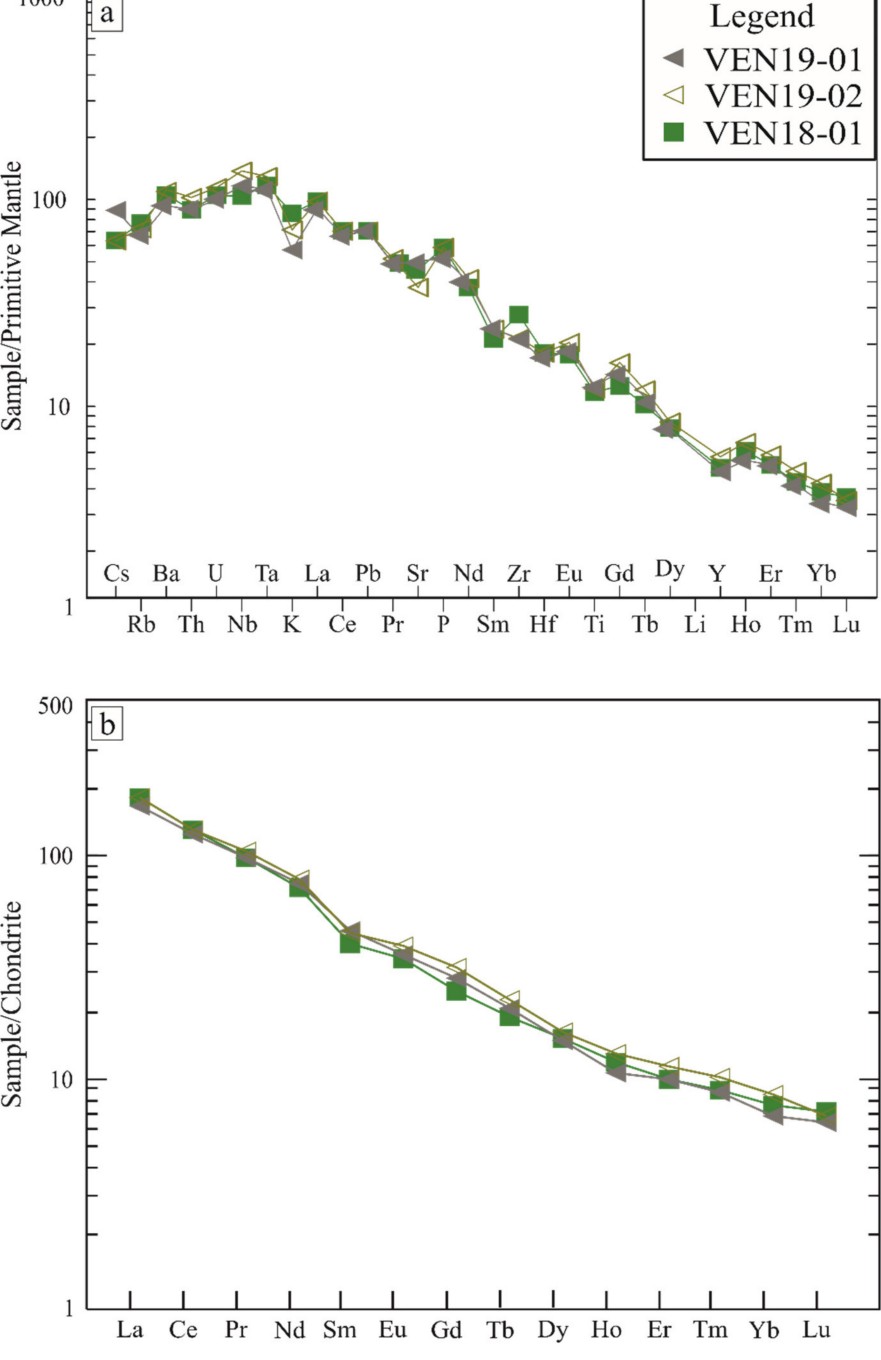

**Figure 11.** (**a**) Primitive mantle normalised spider diagram [23], for the Castelletto and Tonezza del Cimone rocks; (**b**) chondrite normalised REE patterns [34].

The binary diagram of $Al_2O_3$ and CaO proposed by Foley et al. [35] distinguishes the fields of lamprophyres, lamproites and kimberlites. Rock [36] proposed a chemical classification of lamprophyres using the TAS diagram. Based on $K_2O$ vs. $SiO_2$, Raeisi et al. [37] used a TAS diagram adapted to lamprophyres, which may be considered hydrated/carbonated equivalents of more common potassic alkaline rocks. The Tonezza rocks fall in the field of lamprophyres, although the samples from Castelletto di Rotzo (VEN1901 and VEN1902) are in the marginal area between lamprophyres and kimberlites (Figure 12).

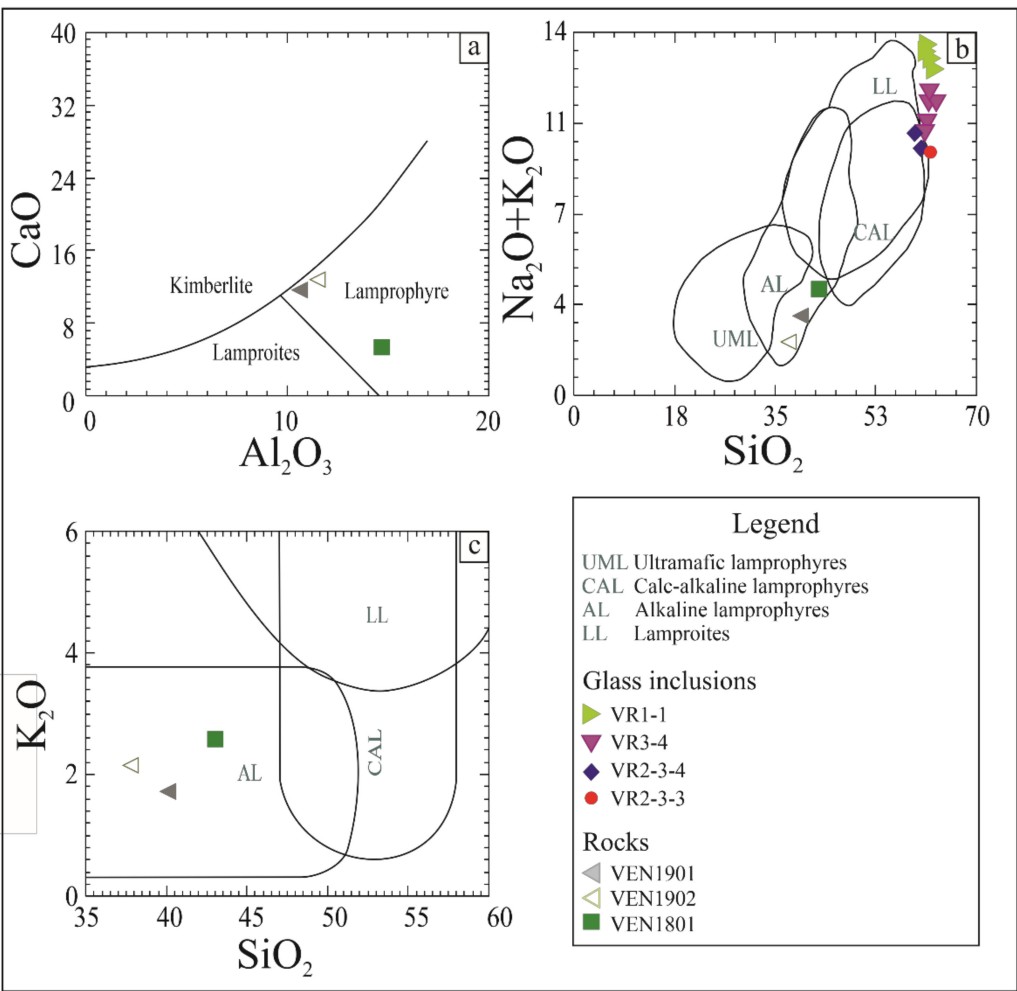

**Figure 12.** Binary diagrams illustrating the composition of the studied rocks and glass inclusions (Table 1). See text for regions. (**a**) CaO vs. $Al_2O_3$ [35]; (**b**) $Na_2O + K_2O$ vs. $SiO_2$ [36]; (**c**) $K_2O$ vs. $SiO_2$ [37].

## 5. Discussion

Zircon forms in various geological processes under conditions that cause considerable difficulties in interpreting their trace-elements distribution. The genetic origin of zircons varies from magmatic, metamorphic/metasomatic, to hydrothermal or a combination of these. Secondary domains quickly form in metamict zircons at low temperatures. A distinction among the above types is based on the elemental partition coefficient (Kd) between either: zircon and crystallising melt in magmatic zircon or between zircon and rocks by sub-solidus phenomena in metamorphic rocks, or by fluid-zircon interaction by hydrothermal systems. In a magma, zircon not only plays an essential role in controlling the abundance and distribution of elements such as Zr (ionic radius 0.87A) and Hf (ionic radius 0.83 A) but can also strongly influence the behaviour of trace elements such as

HREE, Y, Th, U, Nb and Ta. These elements have a small ionic radius (HREEs average 0.99 A, Y 1.02 A, La 1.16 A, Th 1.05 A, U 0.89 A, Ta and Nb 0.64 A) and a high ionic potential. Two characteristics that theoretically make them compatible elements to zircon, in theory. However, geochemical distribution does not systematically fit with the partition coefficient in the zircon lattice, and its trace element content is probably influenced by the crystallisation of competitor HFSE-bearing mineral phases.

Belousova et al. [38,39] used statistics of zircon chemistry from different rocks to generate discrimination factors to attribute their lithological affinity. Magmatic zircons show lower LREE/HREE and higher Th/U ratios compared with metamorphic and hydrothermal zircons. A plot of $(Sm/La)_n$ vs. La (Figure 13a) can be used to distinguish hydrothermal zircons from magmatic zircons, as proposed by Hoskin [40]. Grimes et al. [41] have suggested that the U/Yb vs. Hf may discriminate between zircon from the oceanic crust and continental crust (Figure 13b). Yang et al. [42] suggest that Nb/Hf vs. Th/U may discriminate geodynamic settings.

Vicentine zircons have high $HFSE^{4+}/HFSE^{5+}$ ratios and high Ce; these distributions can be investigated through Harker diagrams, distinguishing genetic origin, magmatotectonic, and source rock type depicted in Figure 13. Magmatic zircon may reflect the original ratio of immobile Nb/Ta pair of their crystallising melt. Figure 13d compares Nb/Ta and La/Ce. Nb/Ta may reflect mantle source compositions where an Nb/Ta $\geq$ 5 reflects a primary mantle composition, whereas low La/Ce may reflect a high $f_{O2}$ in the magma, for example, by an alkaline fluid of hydrated carbonatitic-kimberlitic composition. The Vicentine zircons plot in the igneous field of mantle zircons that characteristically have a high Ta/Nb ratio. Amphibole, garnet and clinopyroxene are competitors as scavengers of Nb rather than Ta, leading to lower Nb/Ta in the co-crystallising zircon. This may explain the low Nb/Ta in zircons from igneous rocks. Alternatively, experimental data show that residual niobate phases in the source favour Ta over Nb, resulting in higher Nb/Ta in the melt separated from or equilibrated with these minerals. This fractionation between Nb and Ta can be ascribed to mantle metasomatism produced by carbonatitic melt and fluids at relatively low temperature [43–45] or by the activity of halogen complexes on Nb and Ta (such as $Na_3TaF_8$ and $Na_2NbF_7$) [46–48], that influence Nb and Ta mobility in fluids and Nb/Ta ratios [49]. Due to the notable absence of F in the zircon inclusions, we deduce that F is possibly hosted by other accessory phases, such as apatite, mica, and amphibole. Whatever was the dominant process, zircons from all three localities plot in the field of magmatic zircons mainly in the anorogenic field (Figure 13c).

Notably, igneous zircons crystallising in oxidizing conditions have positive anomaly of Ce ($Ce^{3+} \geq Ce^{4+}$). Zircons from kimberlites and carbonatites show a minimal or no Eu negative anomaly and show a positive Ce anomaly, like those from the Vicentine area [38]. An estimation of the Ce anomaly is given by Ce/Ce*, where Ce is the chondrite-normalised Ce concentration and Ce* is the average of the chondrite-normalised La and Pr concentrations (geometrically ($sqrt(La^2 + Pr^2)$). The Eu anomaly is given by Eu/Eu*, where Eu is the chondrite-normalised Eu concentration and Eu* is the average of the chondrite-normalised Sm and Gd concentrations (geometrically ($sqrt(Sm^2 + Gd^2)$) [38]. The Ce/Ce* ratio for Vicentine zircons, one of the highest recorded in zircon, is between 10 and 600, like zircons in syenites, but the Eu/Eu* ratio is between 0.5 and 1, plotting in the zircons from kimberlites field (Figure 13e). In Figure 13f, the Vicentini zircons partially overlie zircons from syenites, but most of them have a much lower Y content. In Figure 13g, we reported a distinction for zircons coming from various igneous lithologies based on $ZrO_2/HfO_2$ and $ZrO_2$ [38,50]. The data were compared with the EMPA data of other occurrences in lamprophyres, from the area of Abu Ruscheid (Egypt), from the kimberlites of the Great Slave Lake (Canada) and finally from granitic pegmatites of Laoshan (China) [21,50,51]. The zircons from pegmatites that represent an extreme granite differentiate have higher $HfO_2$ and lower $ZrO_2/HfO_2$.

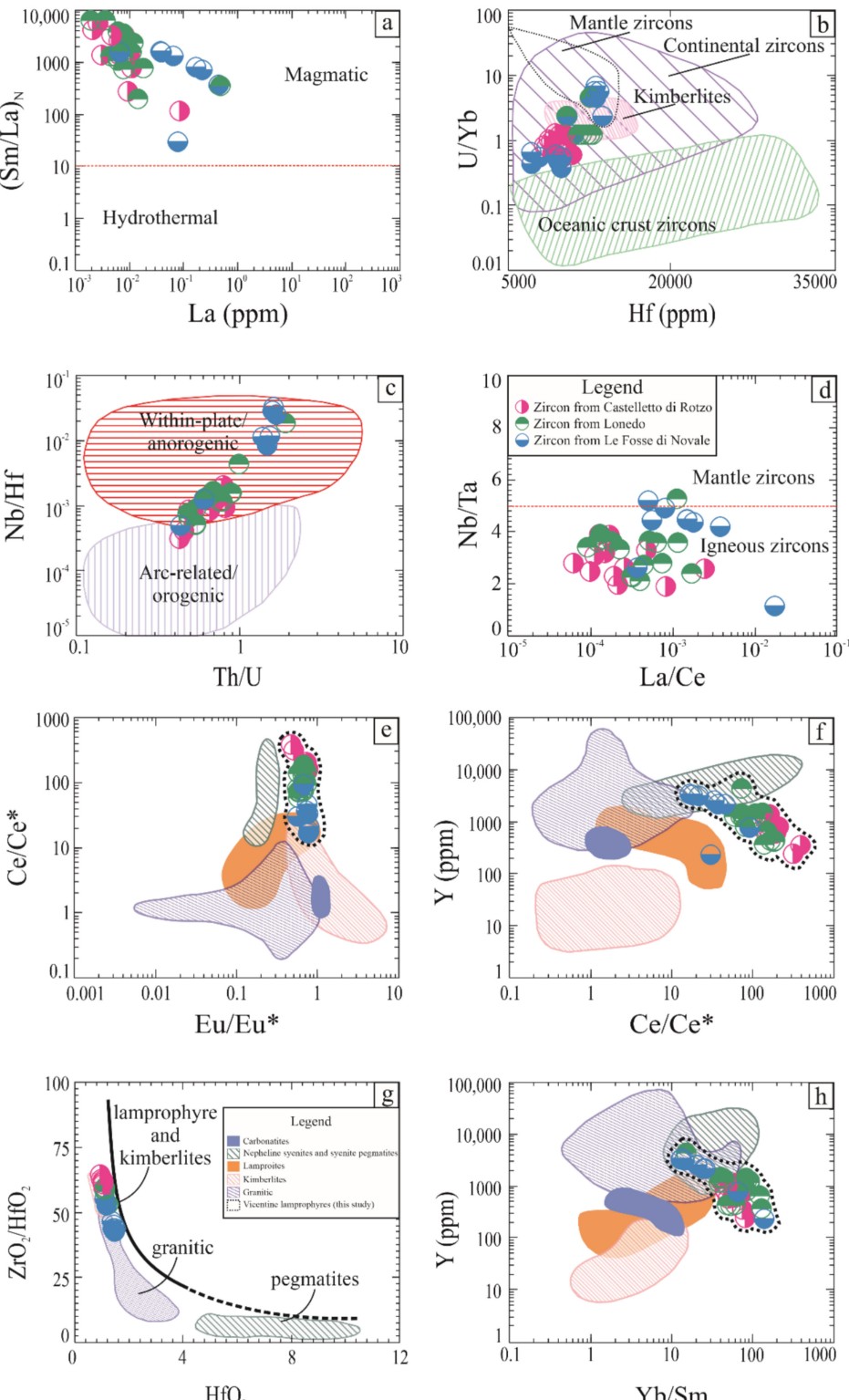

**Figure 13.** Geodynamic classification of zircons from the study area. (**a**) REE ratios separate hydrothermal and magmatic zircons following [40]. (**b**) U/Yb ratio vs. Hf (ppm) indicates tectonic setting such as oceanic and continental crust, and host rock such as kimberlites and mantle zircons, following [41]. (**c**) Nb/Hf vs. Th/U ratio indicates orogenic or anorogenic setting for zircons, following [42]. (**d**) Nb/Ta vs La/Ce ratio indicate differences between the mantle and igneous zircons [43]; (**e–h**) Composition of zircons from different types of rocks about their geochemistry. The curve in Figure (**c**) represents the theoretical fractionation of Zr/Hf in zircon. In addition, the plotting area of zircons in the mantle, lamprophyre, kimberlites, granitoid and pegmatites ([38,50], modified figure).

From a petrologic point of view, inclusions are an additional key to define the nature of the trapped crystallising melt at the time of the zircon crystallisation. Vicentine zircon composition and nature of inclusions suggest they crystallised by a liquid whose parental melt derived by a source metasomatised by carbonatitic–kimberlitic liquids. $CO_2$-rich mantle metasomatism can produce, at low melting degrees, ultramafic alkaline lamprophyres and carbonatitic lamprophyres, as well as documented elsewhere in the Apulian plate [52]. Italian alkaline lamprophyres are often associated with carbonatitic variants [25]. The increase in the mantle volume due to the introduction of metasomatic fluids producing phlogopite-amphibole peridotite may expand the mantle producing an upwelling of it and crustal thinning, as suggested by Wang et al. [53]. A low partial melting degree of such a source is needed to form carbonate-bearing ultramafic alkaline melts rapidly evolving towards a more felsic composition. The presence of glauberite, calcite, and Ba-Ca-carbonate, suggest high-LILE sulphate-rich carbonate melt trapped together with silicate liquid during zircon crystallisation confirms the role of an alkaline carbonate ultramafic melt. The Vicentine zircons confirm this suggestion, plotting in the primitive igneous rock fields of lamprophyres and kimberlites. However, the geology of the Vicentine area does not host outcrops of kimberlitic rocks, and the mantle xenoliths appear to contain no zircons. So, we conclude that the most likely source of the Vicentini soils zircons is the lamprophyritic rock with a carbonatitic affinity.

Zircons of Castelletto di Rotzo are hosted in a calcite–syenite, representing differentiate/cumulate of alkaline lamprophyres whose glasses have a similar composition. The lamprophyre is commonly an alkaline silica undersaturated rock in which baddleyte rather than zircon are more stable as mineral paragenesis. The baddeleyite included in zircons indicates that they are already presented in the crystallising melts, and the zircon crystallised later when the parental melt evolved towards more felsic composition. Figure 13h compared Y vs. Yb/Sm ratio. Again, there is a partial overlap of the Vicentini zircons with the field of zircons from syenites. In general, however, Vicentine zircons exhibit a higher Yb/Sm ratio.

The zircons show a range of age, covering a range of about 8 Ma, less when including uncurtains. To compare Castelletto di Rotzo and Le Fosse di Novale zircons ages from this work and Visonà et al. [4] and data is shown as a box plot (Figure 14). As a result of this comparison, there is no way to make a consistent estimation for Veneto zircons. Lonedo zircons ages are not plotted due to the lack of published data for comparison. The Novale zircons age data are less dispersed for those of Castelletto di Rotzo, but in any case, they are not self-consistent from a statistical point of view. In the combined box plot for Le Fosse di Novale (Figure 14c), an outlier is present, indicating that there is a large dispersion of data. So, we have only a qualitative indication of an age interval between 48.4 ± 1.3 Ma for Le Fosse di Novale and 40.3 ± 3.2 Ma for Castelletto di Rotzo. Median lines are not at the centre of boxes (Figure 14). Skewness indicates that data distributions do not follow a normal distribution. For example, we examine the most discordant data from Castelletto di Rotzo, combining data from this study and Visonà et al. [4]. Despite the small number of datapoints (12 in this study and 8 in Visonà et al. [4]), the Kernel density analysis (Figure 15) shows that our data have a gaussian distribution and Visonà et al. [4] are bimodal. Bimodality may be interpreted as the sampling of different layers. However, for detrital zircons one must use hundreds analysis, but they are costly and time consuming. Therefore, it is not routinely performed. We conclude that the age estimation for Veneto zircons is just indicative of a period from 40 to 48 Ma (Figure 14). As a whole, our Castelleto di Rotzo dating is consistent with the average age of the zircons in the alluvial deposits.

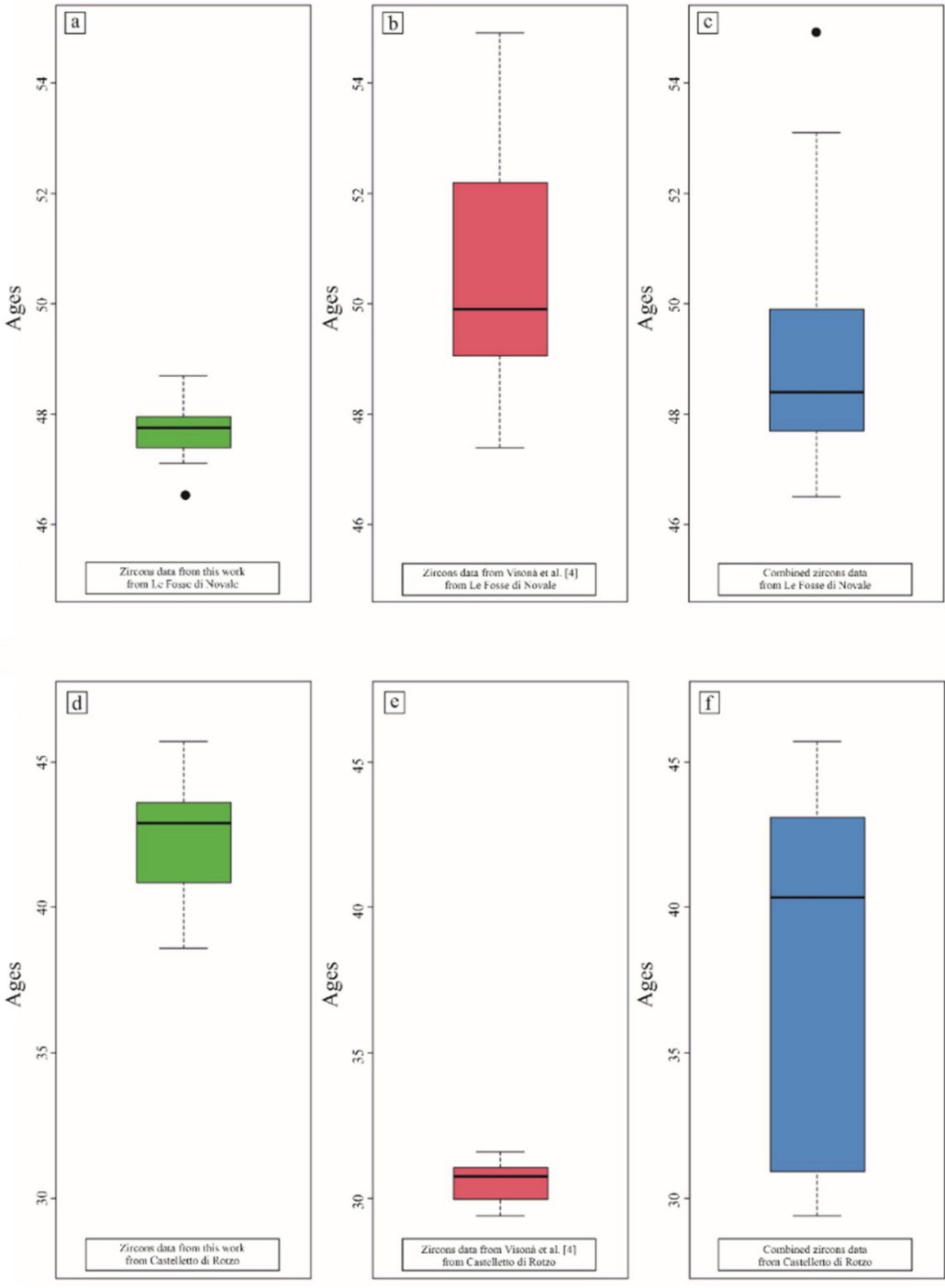

**Figure 14.** Box plot visualisation for U–Pb data. (**a**–**c**) Data from this work, data from [4] and combined data for zircons from Le Fosse di Novale. (**d**–**f**) Data from this work, data from [4], and combined data for zircons from Castelletto di Rotzo. The box plot allows visualising in a standardised way a dataset [54], based on 5- numbers summary: minimum and maximum, that represent the lowest and the highest data point, respectively, excluding outliers; median, represented by the black line in the box, equals to the median value of the dataset; first quartile (25th percentile) equals to the median of the lower half of the dataset (the bottom line of the box) and third quartile (75th percentile) equals to the median of the higher half of the dataset (the top line of the box).

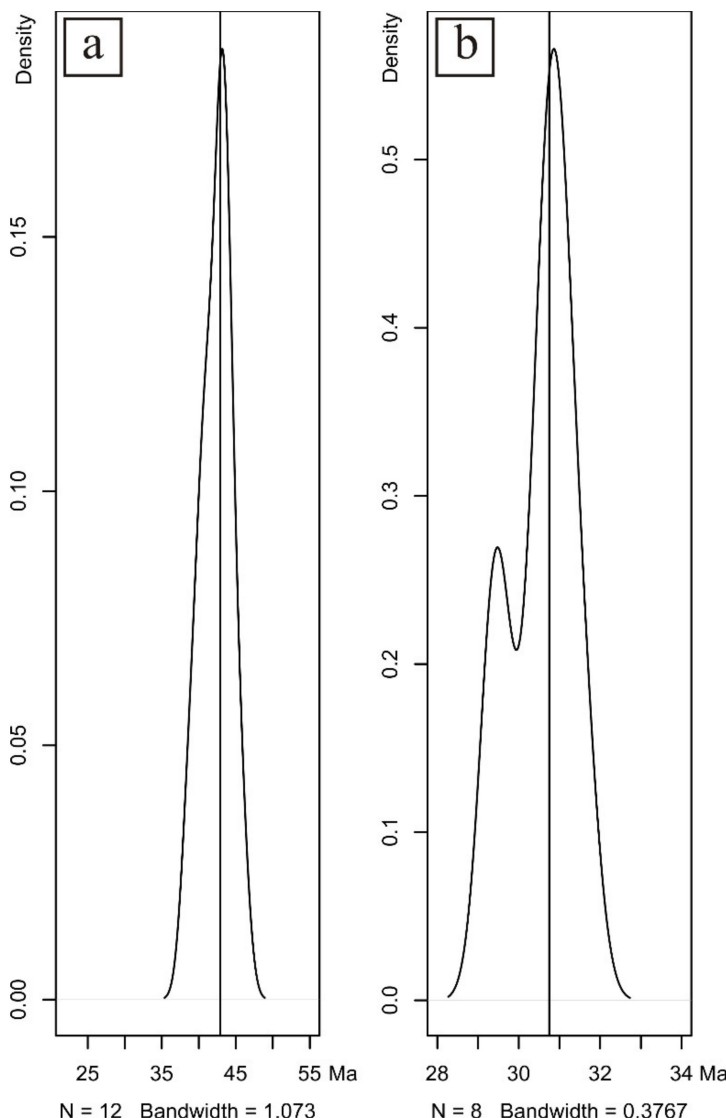

**Figure 15.** Kernel density distribution for U–Pb data. (**a**) Data from this work, (**b**) data from Visonà et al. [4] from Castelletto di Rotzo.

So far, there is no possibility of attributing the provenance of these zircons based only on the ages of the zircons. In fact, mafic and ultramafic rocks with lamprophyric affinity are widespread in the area and may host zircons [4]. Then, magmatic activity can cover a long-lasting time and requires accurate dating. Furthermore, geochemistry, inclusions and mineralogy suggest a general link between zircons and these rocks. Our evidence is not definitive, but we note that the zircons from Castelletto di Rotzo are not only found in the lamprophyre itself but calcite–syenite xenoliths, genetically connected with the lamprophyres and representing a connate cumulate from lamprophyres. There are two possible hypotheses to explain the calcite–syenite. First, carbonate (calcite) was possibly introduced with the zirconium-REE-bearing metasomatic fluids and replaced albite and quartz in the crustal granitic rocks, and thus produced calcite–syenite [55]. In fact, composite inclusions in zircons from Le Fosse di Novale and Lonedo testify for a ZrO–BaO–CaO–SrO and S, $CO_2$-rich fluid or melts that may resemble typical agents of fenitisation. Second, calcite–syenites and lamprophyres may be compatible with progressive fractionation of parental carbonate-bearing lamprophyres to produce calcite–syenite [56]. Alternatively, these inclusions may be linked to the connate lithics at Castelletto di Rotzo, suggesting a common origin from ultra-alkaline fluids or melts. Glass inclusions in the Le Fosse Di Novale rock (Table 1) have a composition akin to syenites. Finally, the average

age of the Castelletto di Rotzo zircons approach that of the zircons in the soils estimated by Visonà et al. [4] and the present study suggesting that they are all derived from the same genetic event.

## 6. Conclusions

1.  The geochemistry of zircons and associated host rocks together testify for emplacement in an anorogenic setting, related to mantle upwelling, responsible for Middle Eocene-Miocene volcanism in Veneto. Mantle upwelling is possibly related to carbonatitic metasomatism, allowing low degree partial melting of the mantle and lamprophyre magma genesis.
2.  Zircon inclusions contain Zr, REE, Ca, $CO_2$, and Ba rich crystallising melts. Furthermore, the glass inclusions in zircon indicate that they are related to felsic differentiation of the mafic primitive lamprophyre melt, and this is confirmed by the presence of zircon in calcite bearing felsic connate lithics.
3.  Origin of zircon-bearing calcite–syenite lithics could be metasomatic (fenite) or magmatic, but zircon geochemistry confirms a magmatic origin.
4.  U–Pb data, carefully examined by statistical analysis, reveals not enough precise and accurate to be specifically related to a single lamprophyre outcrop. However, the dating range corresponds to the Middle Eocene lamprophyre magmatic phase, which is the only igneous manifestation at the time established by zircon.
5.  Zircons from the study area could be a potential source of zirconium element ore. The exploitation of zircon and associated minerals could be of economic value, increasing Italy's national income. The economic potential of zircon in the study area could be improved by extracting other materials such as REE, Hf and Y as by-products. Detailed investigation of the study area and the surrounding areas are highly needed to determine the promising candidate areas of zircon occurrence. Feasibility studies of the zircon exploitability in the study area require further investigation to determine the exact amounts of the zircon ore at depth.

**Supplementary Materials:** The following are available online at https://www.mdpi.com/article/10.3390/min11101081/s1.

**Author Contributions:** Conceptualisation, F.S.; methodology, D.Z. and F.S.; software, N.V.; validation, D.Z., N.V., M.G.P., G.R., V.V.S., E.H.-W., W.B. and F.S.; formal analysis, D.Z., G.R., V.V.S., E.H.-W. and W.B.; resources, D.Z.; data curation, D.Z. and M.G.P.; writing—original draft preparation, D.Z., G.R. and F.S.; writing—review and editing, D.Z., N.V., M.G.P., G.R., V.V.S., E.H.-W., W.B. and F.S.; visualisation, D.Z., N.V., M.G.P., G.R., V.V.S., E.H.-W., W.B. and F.S.; supervision, F.S.; project administration. F.S.; funding acquisition, F.S. All authors have read and agreed to the published version of the manuscript.

**Funding:** This research was funded by FS (HiTech AlkCarb) European Union Horizon 2020 project grant agreement number 689909. Petrography of multiphase inclusions in zircon was done on state assignment of IGM SB RAS (0330-2016-0005). In addition, the Russian Science Foundation partially supported the scanning microscope studies of the zircon-hosted inclusions from the Le Fosse di Novale soils (grant 19-77-10004).

**Data Availability Statement:** Not applicable.

**Acknowledgments:** We thanks Matteo Boscardin for giving zircon samples from Castelletto in collaboration with the Zannato Museum of Montecchio Maggiore (Vicenza). We are grateful to Antonio Canale for his kind guidance in the geological survey in the study area and for co-tutoring a degree thesis on the Vicentine igneous rocks. Finally, we thank Francesca Castorina for suggesting and revising the isotopic data and Fiorenzo Stoppa for statistical data suggestion and processing.

**Conflicts of Interest:** The authors declare no conflict of interest.

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
