# Peer review of "Lamprophyre as the Source of Zircon in the Veneto Region, Italy"

_minerals, doi:10.3390/min11101081_

Round 1

Reviewer 1 Report

“Lamprophyre as the source of zircon in heavy mineral sands (Veneto Region, Italy)” by Daria Zaccaria and co-authors.

BRIEF SUMMARY

In this manuscript entitled “Lamprophyre as the source of zircon in heavy mineral sands (Veneto Region, Italy)” Zaccaria and co-authors present a thorough multi-technique methodology investigating the origin of zircons in 2 heavy mineral sand (HMS) deposits; Le Fosse di Novale and Lonedo both located in the Veneto Region of Italy. The approach is based on an very detailed investigation of the HMS zircons through morphological and inclusion studies as well as U-Pb geochronology and trace-element geochemistry. In an attempt to identify a possible source of the HMS the authors also provide K-Ar dating as well as petrographic and geochemical information of 2 lamprophyre localities; Castelleto di Rotzo and Tonezza del Cimone both in the vicinity of the HMS deposits. From the various datasets acquired the authors discuss the potential rock-type affinities of the zircons from Lonedo (HMS), Le Fosse di Novale (HMS) and Castelletto di Rotzo (Lamprophyre – but actually syenite xenolith in lamprophyre). The authors follow on discussing the geochronology dataset (K-Ar  and zircon U-Pb), defining precise ages for each localities or rock-types . The authors also discuss the nature and the geochemistry of the rocks from Castelleto di Rotzo and Tonessa del Cimone. Finally, the authors attempt to link the zircons from HMS deposits to the ultramafic lamprophyres combining and comparing the geochronology and the geochemical affinities of each localities.

GENERAL COMMENTS

The manuscript really benefits from a very wide range of datasets. It is very pleasant to see that the petrographic and mineralogical observations of both rocks and zircons were not neglected and are carrying essential roles in the attempted interpretations when combined to petrology and more specialist methodologies such as geochronology and geochemistry. The concept of comparing rocks in place and upstream of the HMS is very sound and justified.

However, there are two major flaws that really impact this manuscript.

The first one, which is more of an analytical problem, is the very small number of zircons dated and analysed from each HMS deposits. Provenance studies of detrital zircons usually rely on at least 60 to 100 zircons per site to bring statistical robustness. Here Lonedo is characterised by 3 zircons and Le Fosse di Novale by 13 zricons. They are indeed more U-Pb data points (Lonedo n=16; Le Fosse di Novale n=8), but multiple repeats on individual grains do not really help in the case of detrital studies. I understand that it may be costly both in terms of time and money to analyse a lot of samples, but here at least for the ICP-MS work I would have expected more analyses, the technique is fast and zircons must not have been scarce considering this is heavy mineral sands and that the authors even suggested a possible economic potential for zircon in these sands.

The second major and main flaw, is the fact that instead of extracting and analysing zircons from the Castelletto di Rotzo lamprophyre, which is nicely studied in terms of petrography, petrology, geochemistry and K-Ar dating, the authors extracted zircons from a calcite-syenite xenolith carried by the lamprophyre. Not only they do not provide petrographic, geochemical nor K-Ar of the syenite, but they also do not mention that the zircons are not from the lamprophyre itself until the end of the manuscript (line 672 out of 721!). There is a very brief confusing mention of syenite a little bit earlier (line 636) but no words about a xenolith. This, I believe, generates a lot of confusion for the author throughout the manuscript and particularly in the discussion. For example the authors seem to be contradicting themselves at multiple times within the discussion + conclusion regarding the cogenetic link between the HMS zircons and the Lamprophyre and/or syenite (see detailed comments), which is worrying because the title of the manuscript seems pretty adamant that the lamprophyre is the source of the HMS.

Another example particularly disappointing still linked to the syenite issue, was to not present a proper comparison between the K-Ar ages (which are not really presented and poorly utilised outside of a table) obtained from the lamprophyres (≈30 Ma) and the U-Pb ages obtained for the syenitic zircons (≈42 Ma – It is very normal for a xenolith to be older than the host magma). However, the authors lengthily refer to a previous study of the Castelletto di Rotzo lamprophyre by Visona et al., 2007, in which U-Pb ages of a zircon megacryst from the Castelletto di Rotzo lamprophyre are reported at ca. 30 Ma. Despite both K-Ar and U-Pb ages of material directly related to the lamprophyre returning both similar ages (≈30 Ma), the authors surprisingly continue to dispute the 30 Ma as the actual age of the lamprophyre and continue to assume the 42 Ma of the syenite xenolith as more representative of the lamprophyre. Thus bringing a whole lot of confusion when coming to the final conclusion, especially when the zircons from the HMS returned 44.6 Ma (Lonedo) and 44.7 Ma (Le Fosse di Novale). If the “exotic” nature of the syenite had been stated early in the manuscript, the authors would have better recognised and put more focus on the idea that the syenite may not be representative of the main body of rock and maybe not the most suited sample for this study.

A few suggestions to help improve the manuscript:

  • With the amount of techniques involved and 4 localities (+ xenolith), it is hard for the reader to follow which and how many samples have undergone what. I suggest adding a “Sample” section where these basic information will be provided and summarised. More Info regarding the HMS “outcrops” and how much sand has been collected would be welcome too.
  • Introduce the information related to the syenite xenolith early on and provide detailed petrographic information. If geochemistry is avalaible, provide it too.
  • Create a sub-section for geochronology with the U-Pb data but also including a description of the K-Ar age results.
  • The discussion should build on previous work available, particularly when discussing potential sources of HMS and timing of events
  • The discussion needs thorough rewriting because a lot of interpretations/conclusions are reached very hastily and some sections that should belong to the discussion are found in the Results section or even in the Conclusions.
  • Can you discuss the abundance of syenite xenoliths in the lamprophyre?

DETAILED COMMENTS

Abstract

-Line 24: “ingredient” replace with resources

-Line 24: “green-tech” replace with green technologies

Introduction

-Line 30: “in small quantities” replace with as accessory minerals

-Line 31: “concentrated” replace with concentrates. The use of the simple present expresses general truths, repeated actions and is more appropriate here.

-Line 32: “heavy-mineral-sands” doesn’t need to be hyphenated

-Line 33: “often form their own mineral phases as inclusions”

-Line 35: “Plasma-spray technologies” develop this. For example: Zircon has a lot of applications in modern industry. For instance, Plasma-spray...

-Line 35: Add references

-Lines 35-36-37: The whole sentence needs to be reformulated: Yttrium extracted from zircons is used as a strengthening element in many materials and metal alloys.

-Lines 35-36-37: Add references

-Line 38: “hi-tech” it is usually spelled high-tech

-Lines 64-65: “powerful tool for typifying their magmatic source, geological processes, and a preliminary evaluation of their economic potential” replace with powerful tool to identify their magmatic source, constrain geological processes, and serve as a preliminary evaluation of their economic potential.

Geological setting

-Line 86: Add references

-Lines 87-88-89: Add references

-Lines 89-90-91-92: Add references

-Line 122: “ the deeper fresh sands, which may host very high concentrations in some layers” rephrase this sentence I am struggling understanding the meaning here.

-Lines 123-124-125-126: Maybe because of my lack of understanding of the previous sentence, nonetheless I do not see how you come to this conclusion. It would be helpful to add a section describing the in more details the location and morphology (is this a current river bed? Is the soil in place?) of the HMS deposits and also how the sands were sampled, how deep? What I am about to write will have to be taken with a lot of caution because of the extremely small number of detritic zircons forming the dataset presented in this manuscript, but the fact that you seem to only find 1 single age per deposit and no mixing would maybe suggest that the source is nearby.

Methods

-Line 172 and throughout the manuscript: “standard(s)” replace with reference material(s)

-Line 175 to 182: Add laser parameters particularly laser spot size.

-Line 175 to 182: Were  zircon trace-elements and U-Pb acquired simultaneously? Modify text accordingly.

-Line 175 to 182: Provide session averages and accepted values for Plesovice and 91500 reference materials.

-Line 175 to 182: How was the LA-ICP-MS data reduced? In-house excel spreadsheet, Glitter® Iolite® other? Please add to text with necessary references if needed.

 -General comment: The Methods section lacks information regarding the analytical procedure for the whole-rock geochemistry. Which technique was used (XRF? ICP-MS/OES?)? Please add.

Results

-Line 237: Caption Figure 5. – Crystal b: In zircons cleavages are poor/indistinct. Could this rather be fracturing? Maybe along growth planes?

Also are these 3 zircons representative of each locality in terms of morphology and CL patterns?

-Line 283: “ZrO2 and F are below detection limits (<0.005 wt.%).” This seems to be an extremely low detection limit for an EDS system. Even if calibrated with reference materials.

-Tables 3-4-5 LA-ICP-MS TE zircons:

It would be very good to report uncertainties in these tables.

If analysed reporting concentrations of P, S would be interesting to show potential contribution from inclusions such as apatite or sulfide. Y would also have be interesting to have as it is used to constrain the petrogenesis of zircon (e.g. Pupin, 2000; Belousova et al., 2002). Actually after reading the manuscript further Y seemed to have been analysed and utilised. Same with Hf (and a few other elements- see Figure 11 a). Why are they not available in these tables???

You need to identify in the table which ones are in wt% and which ones are in ppm. Also nowadays the International Association of Geoanalysts (IAG) strongly recommends the use of µg g-1 (or µg/g) for trace element concentration by LA-ICP-MS (or other solid state techniques) and µg ml-1 for solution work (http://www.geoanalyst.org/say-no-to-ppm).

-Lines 342-346-350: “zircons analyses”

-Line 355: “HFSE are note easily incorporated in zircon”

-Line 360: “anomaly for Ta” replace with anomaly relative to Ta

-Line 361: “are much enriched in some HFSE, such as Nb, Ta, La, and Ti”.

-Lines 362-363-364: This is absolutely correct, however the main message figure 11b is conveying is that the zircons from Le Fosse Di Novale are overall less depleted in REE, and particularly in LREE, with presumably a higher La/Sm ratio

-Line 372: I would create here a sub-section for the zircon geochronology e.g. 4.4 Zircon U-Pb geochronology

-Line 373: “[22].” Full stop missing

-Line 374-375: “42.36 Ma….44.57 Ma….47.70 Ma” Please add uncertainties.

-Line 376: “Our data are different from those by Visonà, 2007 referring to the zircon from Castelletto di Rotzo.” replace with of the zircon from Our data are different from those of Castelletto di Rotzo analysed by Visona et al. [2].

-Line 377: “Visona, 2007” replace with Visona et al [2]

-Line 377: Comment on “smooth and rounded at the edges” Visona et al [2] give very little morphological description of their zircons. They only briefly show subhedral to subrounded images of representative zircon megacrysts of the VVP (fig2) at the beginning of their publication. I am assuming that your statement “smooth and rounded at the edges” comes from the CL images (fig 4. Visona et al).  I would be extremely cautious interpreting zircon morphology from CL images (fig 4. Visona et al). Zircons are mounted in epoxy resin and the shape of the zircon's surface exposed is highly dependent on the amount of polishing. I have seen perfectly euhedral grains appear fully rounded, only because the samples had not been polished down enough and only  small fractions of the grains were exposed.

-Line 378: “Luminescent areas” replace with Areas with a brighter CL response

-Line 386: Comment: I may have missed it but I didn't find any zircon ages for Lonedo in Visona et al., 2007

-Lines 386-387: “Visona, 2007” replace with Visona et al [2]

-Table 6 - Results of U-Pb age calculation:

  • The uncertainties on the 207Pb/235U are quite high for approximately half of the dataset (especially analyses of Castelletto di Rotzo zircons).
  • 12 zircons analysed for Lonedo HMS and 8 zircons analysed for Novale HMS. THIS IS NOT ENOUGH. Detrital and provenance studies require at least 60 to 100 grains if not more to provide statistical significance and help get away from the human bias in selecting the zircons to analyse. Nature has already separated and concentrated the zircons in these HMS, so I am sure this was not due to a lack of zircons. This bring a true limitation to this study.

-Figure 12: Comment: Try keeping the exact same color scheme between figures. I would personally take the colour scheme from this figure (which has more contrasting colours) and apply it to all the trace-element diagrams. It will greatly help the reader.

-Line 398: Comment on the 4.3 Petrography section: There are 2 rocks that appears later in the manuscript that are not described in this section, but should be. You present some WR data (K-Ar ages and trace elements) for Tonessa del Cimone. You also very lately revealed that the zircons extracted from the Castelletto di Rotzo sample are not from the lamprophyre described in this section but from a syenite xenolith within the lamprophyre. A petrographic description of this syenite would be critical to help the discussion relative to the origin of such xenolith.

-Line 401: “[23, 24].” Full stop missing

-Lines 401-402-403: “ Therefore, the Subcommittee on Igneous Rock Systematics of IUGS updated recommendation suggests the classification of lamprophyres, a scheme proposed by [25]” rephrase -> complex and unclear.

Also, if explicitly calling a publication in the text, the authors have to be spelled out. E.g. "showed by Author et al., [25]" This is a recurring problem in the entire manuscript.

-Line 404: “…in this scheme but it is essentially…”

-Table 6 Lamprophyre classification: This is the second table #6. Correct the number of this table and following ones accordingly, as well as in text referencing.

-Line 412: “zircon age (Table 7)” referencing wrong table or wrong age type (K-Ar)

-Line 413:”[2]” replace with Visona et al. [2]

-Line 414: “Tonezza dikes” no petrographic description provided for these rocks. Any zircons available?

-Line 418: “Table 7” replace with Table 8

-Table 7: Results of K-Ar age calculation: Why is this table not described in the Results section?

-Line 420: “is due to [26] and [14]” replace with was made by Ogniben [26] and De Vecchi [14]

-Line 421: “were published  by [27,28]” replace with were published by Boscardin et al. [27, 28] or replace with Further studies with more mineralogical data were also later published [27, 28].

-Lines 421-422: “[14] classified the rock” replace with De Vecchi [14] classified the rock

-Line 422: “More recently, [3], published” replace with More recently, Zorzi et al. [3], published. I think that I made my point clear, I will for now on just refer this kind of problems with Authors

-Line 425: “The rock” comment: which one? From the the dyke, the diatreme or both? Or do you mean your sample? From which part is your sample coming from?

-Line 426: remove ”disaggregated”. Xenocrysts is self-explanatory.

-Line 447: “dyke clasts (autoclasts)” Comment: not clear. Are you talking about dyke clasts in the diatreme?

-Line 448: “They represent high-temperature subvolcanic lapilli produced by 448 breccia fluidisation operated by CO2-H2O-rich fluids.” Interpretation. Should be in discussion part, unless you have supporting references for these specific rocks, which should then be cited here.

-Line 473: Authors

-Lines 473-474: You haven't presented rock geochemistry yet. This entire paragraph is very interpretative and might suit better in the discussion than in the petrographic description.

-Line 500: Missing numbering of the sub-section header. Should be 4.4 or 4.5 if you create a sub-section for zircon U-Pb geochronology

-Table 8: Table 9. Here again, it is usually good practice to present associated uncertainties

-Line 515: Missing reference to  (Fig. 15) after “R1-R2 diagram”

-Line 516: “EMPA” the zircon inclusions were studied with SEM-EDS not EMPA.

-Line 518: Remove “Fig. 15” it needs to be called earlier at line 515

-Line 529: Missing reference to  (Fig. 16a) after “multielement diagram”

-Line 532: “REEcn” and “La/Lucn” replace with REEN and (La/Lu)N

-Line 532: Missing reference to  (Fig. 16b) after “patterns”

-Line 532: “for both the rocks” replace with for all the rocks. Unless both Castelletto di Rotzo analyses (VEN-1901 & VEN-1902) are duplicates or from the same sample. But here again, if this had been stated earlier in the methodology section or in “a sample section”, this would be much clearer.

-Line 532-533: “ There is no Eu anomaly, usually seen in rocks without plagioclase.” Comment: Slightly simplistic statement. It is relatively common for ultramafic magmas to not have negative Eu anomalies. Eu is generally not fractionated in the deep source of such magmas. Negative Eu anomalies can occur but, , for example, would require a stage of fractional crystallisation of plagioclase along the way.

-Line 541: Authors

-Line 542: Authors

-Line 545: “samples VEN1901 and VEN1902” replace with samples from from Castelletto di Rotzo (VEN1901 and VEN1902). Comment: Location often speaks more to the reader who isn't familiar with your work than sample codes. Also fix inconsistency regarding sample names/labels between tables, figures and text: VEN1901 vs VEN19-01

-Line 552: Missing reference to  (Fig. 18) after “chemo-tectonic diagrams”

-Line 554: Remove “Fig. 18” it needs to be called earlier at line 552

Discussion

-Line 563: Authors. Belousova et al., 2002 [50] more appropriate here

-Line 579: “Hf vs U/Yb” replace with U/Yb vs Hf

-Line 579: Authors

-Line 580: Authors

-Lines 585-586-587: “ CO2-rich mantle metasomatism can produce, at low melting degrees, ultramafic alkaline lamprophyres and carbonatitic lamprophyres that host the most of Vicentine zircons plot in the igneous field preserve high Nb/Ta ratio, like mantle zircons.” Please rephrase (particularly the highlighted part). Difficult to understand the meaning of this sentence.

-Lines 594-595-596: Comment: No fluorine in the zircon doesn't mean that F wasn't in the melt. It could have partitioned into another phase in the lamprophyre. Apatite, mica and amphibole will be potential host for F.

-Line 619: “approximating” comment: Even if some zircons seems to be close to the kimberlite field in terms of Y concentrations, the lowest are around 300 ppm and the highest at around 3000 ppm. As you said kimberlites have Y < 100 ppm.

-Lines 622-623: Not sure what you are referring to here. Are you referring to Figure 20 in general. if so this should at the beginning of this specific section. I am also guessing you are referring to [50, 51] and not just [51]

-Lines 636-637: “ Zircons of Castelletto di Rotzo are hosted in a calcite syenite” Comment: I am very confused now. I thought the Castelleto zircons where extracted from the Castelletto lamprophyre that you have petrographically studied at length in this manuscript. Are they from a syenitic xenolith in the lamprophyre?? Syenite is a coarse-grained intrusive rock, petrographically and mineralogically different from a lamprophyre. I really recommend you add a "Sample" section early on in the manuscript in which you precisely describe how many rocks from each locality and which samples have undergone this or that technique.

Alright, now I have read the very end of the manuscript and you state the syenite xenolith. This should be stated at the very beginning of the manuscript, in the Sample section I recommend. The fact that this is a xenolith obviously bring the very critical question of relevance of these zircons to date the Castelleto di Rotzo lamprophyre and all subsequent interpretations/conclusions related to the origin of the zircons in the HMS studied.

-Line 637:  “ a late differentiate (or cumulate)” Comment: or early differentiate or something picked-up along the way (quite a common thing for xenolith, especially if U-Pb returns an older age than the K-Ar of the lamprophyre itself). See previous comment regarding the syenite issue.

-Figure 20: Is the lamprophyre field displayed on the diagrams defined by this study only? if yes, I would amend the legend to "Venetian lamprophyres (this study)" unless you can compile a larger set of data from lamprophyre worldwide. Other option is to remove this lamprophyre field all together and just keep your data points.

-Line 648: Caption Figure 20 – replace “[51]” with [50, 51]

-Line 650-651: “42.36 Ma, which differs from the estimated U-Pb age of [2] for the Catelletto di Rotzo zircons” Comment: Most likely because you analysed zircons from a syenite xenolith and not from the lamprophyre. While Visona et al have analysed zircon megacrysts that they assumed were cogenetic with the lamprophyre. The K-Ar reported in this study for Castelletto di Rotzo (28.3 ± 1.4 Ma) are in agreement with Visona et al. zircon age for Catelletto di Rotzo (29.1 ± 2.0 Ma intercept Tera-Wasserburg).

-Line 651: “[2]” “[2]” Authors

-Line 651-652: “However, [2] attributes the younger age of these zircons to an excess of Pb.”

Comment: No they didn't. they did remove a couple of analytical spots from 2 samples (ATD and SP04). The spots removed were giving slightly younger ages because of high common-Pb contamination. However these sample were not from Castelletto di Rotzo. In the contrary the data point associated with the Castelletto di Rotzo  zircon (MB186) were showing very low level of common-Pb and  therefore the age derived from this zircon is analytically very robust. It is very disappointing that you desperately try to associate the older age of the xenolith (42.4 Ma) to the lamprophyre when both whole-rock K-Ar and Visona previous work are in agreement for a much younger age (≈ 29 Ma). Even if the analytical point mentioned, which is by the way analytically inaccurate and also not true for the sample in question (sorry to tell, but very poor cherry picking here), was even the slightest  true, this is a 13 Ma difference, this is very consequent. Dating zircons from a xenolith to date a lava is the massive flaw of this study.

-Lines 652-653: “the K-Ar age of the lamprophyres, fails any statistical test compared with the age estimated by [2]” Comment: As I mentioned above…. Absolutely NOT correct!!! the K-Ar age you presented  for Castelletto di Rotzo is 28.3± 1.4 Ma and the age obtained by Visona et al  for Castelletto di Rotzo (M186) is 30.5 ± 0.4 Ma (weighted mean) and 29.1±2 Ma (Tera-wasserburg intercept). Seems in accordance to me. Much more so than the 42.4 Ma that you obtained with the zircons in this study. Which makes sense now that I know the zircons were from a syenite xenolith and not the lamprophyre itself. Here you have been dating 2 different things the K-Ar was dating the lamprophyre and the zircon U-Pb dated the crystallisation of the syenite. 

On a positive note the Novale ages for this study and Visona et al seems to agree. This study 47.6 Ma, Visona et al., (N5) is 49.1 ± 1.0 Ma (weighted mean) and 47.9 ±3.6 Ma (Tera-wasserburg intercept).  I think this is worthy to mention somewhere in your text.

Similarly, the Lonedo age of this study 44.5, resemble the 44.9 ± 2.8 Ma of Visona et al for a porphyritic basanite lava (ATD) belonging to the Paleogene Veneto Volcanic Province in which Lonedo also seats. it would therefore be interesting to compare the trace-element data of ATD with the Lonedo traces from this study.

This could represent an alternative source of origin for Lonedo zircons.

-Lines 653 trough 665: I will not go into too much detail for this part because the d,e,f part of the diagram is flawed by the fact that you are comparing apples and pears because of the syenite xenolith. it therefore should be removed.

-Lines 670-671-672: “Thus, there is the possibility of a genetic link between lamprophyres and zircons, but our data cannot demonstrate a coeval formation for zircons and lamprophyres. Our evidence is not definitive” Comment: I very much agree with this statement. However the title of the manuscript claims the opposite and so is the statement at line 684 “the age of the Castelletto di Rotzo zircons approach that of the zircons in the HMS estimated by [2] and the present study suggesting they are all derived from the same genetic event”.

-Lines 672-673: “zircons from Castelletto di Rotzo are not found the lamprophyre itself but in calcite-syenite xenoliths” Comment: Everything Makes Finally Sense!!!! You cannot keep this kind of critical information for the end. This has to be stated at the very beginning of the manuscript. (I liked the intrigue, but this is not an Agatha Christie murder novel hahaha)

-Lines 677-678: “composite inclusions in zircons testify for ZrO-BaO-CaO-SrO abd S, CO2-rich fluid or melts that may resemble typical agents of fenitisation” Comment: Although I like the idea of a pre-metasomatic event turning the rock into syenite before being entrapped as a xenolith in the lamprophyre. You haven't described any inclusions for Castelletto zircons. You only have described inclusions in the zircons for HMS (Novale and Lonedo).

-Lines 681-682: “Glass inclusions in the Castelletto di Rotzo rock (table 1)” replace  with Glass inclusions in the La Fosse di Novale (table 1) Comment: Table 1 presents inclusion data for La Fosse di Novale not Castelletto.

-Lines 684-685: “that they are all derived from the same genetic event” Comment: This is a big shortcut. The "syenite (≈42.4Ma)" could be one potential source for maybe Lonedo HMS which gives just a slightly older age (≈44.5Ma), However, from Visona et al. [2] another potential source could be the porphyritic basanite of the VVP (sample ATD at ≈44.9Ma). Zircons from Novale both in this study and Visona et al. are at around ≈48Ma, which seems a bit of a stretch to be related to the Syenite (≈42.4Ma). But considering the large  amount of  intermittent volcanism in the area from the Paleocene to the Miocene, there is quite possibly another source yet to be identified.

However, like stated Line 671 “ our data cannot demonstrate a coeval formation for zircons and lamprophyres”,  it is true that the lamprophyres from Castelleto di Rotzo and Tonessa del Cimone (≈30Ma) doesn't seem to be the magmatogenetic source of these zircons (>> 42 Ma).

-Lines 691-692-693: What's the apparent size of the HMS deposits in Lonedo and Fosse di Novale? It would be good to give a rough estimate at the beginning of the manuscript when you describe the sampling locations.

Conclusions

-Line 695: “5. Conclusions” replace with 6. Conclusions

-Lines 699 through 708: This should have been discussed in the discussion part in more details and a summary added in the conclusions. Conclusions should just summarise preferred interpretations that have been reached in the discussion.

I am not deeply familiar with the geological history of the area, but according to Visona et al. [2] and reference therein, the area was in a tensional setting during the Tertiary with even a syn-volcanic graben. Pointing possibly towards rifting volcanism as another possibility for the emplacement of these rocks. Alkaline and carbonatitic melts are often associated with continental rifting.

-Lines 709 through 715: Same than for the point 2 of the conclusion. You haven't written about this before. Conclusions should just summarise results and points discussed in the discussion. It is not the place for new information. Also I am not entirely sure of the overall relevance of some info given here.

-Lines 716 through 719: I agree! However, there is some age correlation with your lamprophyre dated at ≈30Ma by K-Ar and by U-Pb Visona et al [2] (zircon MB186) also at ≈30 Ma. Nevertheless, it seems like the 2 lamprophyres presented here are actually not the source of the few HMS zircons presented.

Once again, the contradiction reappears, as you are writting that you can't say if the lamprophyres are the source of the zircons, yet the title of the manuscript states the opposite: "Lamprophyre as the source of zircon in heavy mineral sands" as well as line 684 “the age of the Castelletto di Rotzo zircons approach that of the zircons in the HMS estimated by [2] and the present study suggesting they are all derived from the same genetic event”.

Reviewer 2 Report

General comments:

My comments are summarized in the following 4 points.

Figures.

     This paper have too much figures to explain the ‘discussion’ section. A scientific paper should be straightforward.

Methods.

     Your experiments are too much for us to readily understand them, while ‘methods’ section is too simple. In order to help better understanding of your work, please make methods complete. Particularly, this paper has few explanations about data treatments on zircon U–Pb and trace elements analyses and whole-rock geochemistry. I cannot judge whether or not your interpretations are based on your data.

Discussion-1.

     A lamprophyre is assumed to include zircons in this paper. However, the lamprophyre is commonly an alkaline silica-undersaturated rock, which indicates that baddeleyites rather than zircons are stable as mineral paragenesis. For the geochemical discrimination, please add the explanation about the crystallization process of zircons in lamprophyric magmas.

Discussion-2.

Age interpretation is insufficient. At first, please make U–Pb age relationships between your and previous works more explicit. At lines 651–652, you stated that [2] attributes the younger age of these zircons to an excess of Pb, but excess of Pb (initial Pb?) lead to older ages. In this case, a data treatment on initial Pb correction are more important. Please re-evaluate your U–Pb data. At second, age relationships between U–Pb and K–Ar data were not enough discussed here. You need to emphasize the age correlations between HMS minerals and host rocks.

Specific comments:

Line 32. What does this elemental order follow?

Lines 33–35. I am not familiar with such an industrial technique. Please add related references.

Line 137. What types of alteration dose it mean? Chemical? Color?? If this is chemical alteration, the following metamorphism is inappropriate due to its open system.

Lines 199–200. Zircon coloring is not of radiogenic origin. This will partly result from the metamictisation.

Lines 203–205. I cannot understand evidences that zircon coloring related to the trace concentrations. If you have substantial reasons, please add them.

Lines 231–232. There is controversy about the nature of CL emissions. Its interpretation for natural zircons are not easy. Why do you compare measured chemical compositions of zircons with the CL features?

Lines 354–356. Pb4+ is not abundant in natural magma. In addition, expression ‘large-radius HFSE’ is less familiar with me. Please add concrete examples.

Lines 354–363 and Figure 11. A ‘anomaly’ is inappropriate for the explanation of Figure 11. In order to define ‘anomaly’, an estimate for the standard composition is required. A spider diagram of Figure 11 should be applied for bulk rock compositions. I think that Zr, Hf, La, Ti, Nb, and Ta abundances are non-anomaly.

Figure 1 caption. Explanations for caption ‘a’ and the whole map are not included here. Please insert.

Figure 12. U-Pb analytical data has systematic bias depending on analytical sessions. For example, Number 1–5 data of Castelletto di Rotzo are younger than number 6-12 data of that. This bias also affect MSWD values. Please explanation this cause.

Figure 19a and 19d. Their discrimination lines depend on only elemental ratios of vertical axes. What does elemental ratios of horizontal axes mean?

Figure 20. Legend does not include blue solid areas. Please add its explanation.

Figure 21. Median lines are not at centers of boxes. This indicates that data distributions does not follow a normal distribution. If so, please clarify what type of a statistical distribution your measurements follow in order to calculate maximum likelihood values and its uncertainty.

Table 1. How do you calculate FeO content? Valence of an iron is variable in a nature.

Table 4. Please make the unity of each elemental abundance clear. In addition, detection of Al, Ca, Ti, V, Fe, Mo, Ba, LREE, and W are not easy in LA-ICP-MS analysis. How do you check their analytical accuracy: detection limit, linearity of a calibration curve, and effects of spectral and non-spectral interferences.

Table 6. Does Rho mean covariance? Agilent 7700 separately monitors targeted elements, which means that data acquisitions are independent. A cause for the covariance is obscure.

Unit. Please express based on SI Unit rules.

Typo:

Dash. a dash is used as the same meaning as ‘to’. This is incorrect. The dash should be replaced by en dash.

Lines 32–33. Ti and Nb

Line 63. also include inclusion data

Line 72. South-Alpine tectonics is still

Line 80. Thermometamorphism

Line 76. and postdates the initial magmatic activity

Line 91. and located north of the more external South-Alpine thrust

Line 121. 3 g/cm3

Table 6. 207Pb/235U and 206Pb/238U

Reviewer 3 Report

The manuscript is devoted to geochemical studies of zircons from the sands of the Veneto Region, Italy. On the basis of comprehensive studies of these zircons and mineral inclusions, the authors draw a conclusion about their magmatic-tectonic genesis.

In my opinion, one of the most important results of these studies is the establishment of a genetic link with carbonate-rich alkaline rocks.

The article makes a good impression. The results fully reveal the intention of the manuscript, and the abundance of illustrations perfectly complement the text part.

The article is written in a good scientific language, and the approaches and methods of studying substance used by the authors correspond to the world level. 

The manuscript is of scientific interest and can be published in the journal,  after a thorough check of references and text design according to the journal requirements.
My remarks:
1. It is advisable to expand the Introduction a little and give more
references to contemporary literature.
2. Check all literature references.

Round 2

Reviewer 1 Report

The manuscript has significantly improved in terms of presentation, clarity, and overall soundness. The discussion is clearer, and the interpretations have been moderated. I am very pleased with the way the authors have handled the suggestions/comments.

It is sad that the lamprophyre K-Ar ages have been removed all together. This kind of data is expensive to obtain both in terms of time and funds and is extremely beneficial to the community. With the mitigated interpretations provided, It would have been possible to integrate this dataset in the discussion without impacting the conclusions.

Regarding the reply: "No published work on this subject analyses a larger number of data than ours". I have no doubt regarding the Vincentine zircons and especially Visona's work. Visona et al. did not analyse enough grains and their ages were statistically poor. However, this study is a detrital zircon study and there are literally countless studies in the literature using this methodology. This type of approach using detrital zircons has been around for several decades, and some clear statistical principles have been put in place, for example, Dodson et al. (1988) states that 59 grains (not single analyses) must be analyzed to achieve 95% confidence of finding every population that exists at the 5% level in a given sample, and some more recent studies suggest a higher number is required. 

Line 296: wrong numbering for Visona et al.

Author Response

REVIEWER 1 ROUND 2

The manuscript has significantly improved in terms of presentation, clarity, and overall soundness. The discussion is clearer, and the interpretations have been moderated. I am very pleased with the way the authors have handled the suggestions/comments.

It is sad that the lamprophyre K-Ar ages have been removed all together. This kind of data is expensive to obtain both in terms of time and funds and is extremely beneficial to the community. With the mitigated interpretations provided, It would have been possible to integrate this dataset in the discussion without impacting the conclusions.

Regarding the reply: "No published work on this subject analyses a larger number of data than ours". I have no doubt regarding the Vincentine zircons and especially Visona's work. Visona et al. did not analyse enough grains and their ages were statistically poor. However, this study is a detrital zircon study and there are literally countless studies in the literature using this methodology. This type of approach using detrital zircons has been around for several decades, and some clear statistical principles have been put in place, for example, Dodson et al. (1988) states that 59 grains (not single analyses) must be analyzed to achieve 95% confidence of finding every population that exists at the 5% level in a given sample, and some more recent studies suggest a higher number is required. 

We thank you for your advice and suggestions, which enabled us to make our manuscript better.

The decision to remove the K-Ar ages was made because these data are beyond the scope of this paper. They will be the subject of a specific work later.

Regarding the study of detrital zircon dating, the number of zircons analysed may vary. You mentioned Dodson et al. (1998) with 59 grains (not single analyses). In other cases, however, the number of analyses is comparable to ours. For example, Xu et al. (2008) use 32 analyses. In any case, we argue that the number of analyses is sufficient for our purpose, which is not so much to have an exact date, but more to make genetic attributions. 

Line 296: wrong numbering for Visona et al.

Done.

Reviewer 2 Report

I recommend the addition of check standard results for U-Pb dating and trace-element analysis. This makes validation of your analytical results.

Author Response

REVIEWER 2 ROUND 2

I recommend the addition of check standard results for U-Pb dating and trace-element analysis. This makes validation of your analytical results.

Check standard results for trace-element analysis are already present in supplementary material 1. To add check standard results for U-Pb dating we have prepared a second supplementary material with the complete data.